# LEARNING TO SOLVE MULTI-ROBOT TASK ALLOCATION WITH A COVARIANT-ATTENTION BASED NEURAL ARCHITECTURE

## ABSTRACT

This paper demonstrates how time-constrained multi-robot task allocation (MRTA) problems can be modeled as a Markov Decision Process (MDP) over graphs, such that approximate solutions can be modeled as a policy using Reinforcement Learning (RL) methods. Inspired by emerging approaches for learning to solve related combinatorial optimization (CO) problems such as multi-traveling salesman (mTSP) problems, a graph neural architecture is conceived in this paper to model the MRTA policy. The generalizability and scalability needs of the complex CO problem presented by MRTA are addressed by innovatively using the concept of Covariant Compositional Networks (CCN) to learn the local structures of graphs. The resulting learning architecture is called Covariant Attention-based Mechanism or CAM, which comprises: 1) an encoder: CCN-based embedding model to represent the task space as learnable feature vectors, 2) a decoder: an attention-based model to facilitate sequential decision outputs, and 3) context: to represent the state of the mission and the robots. To learn the feature vectors, a policy-gradient method is used. The CAM architecture is found to generally outperform a state-of-the-art encoder-decoder method that is purely based on Multi-head Attention (MHA) mechanism in terms of task completion and cost function, when applied to a class of MRTA problems with time deadlines, robot ferry range constraints, and multi-tour allowance. CAM also demonstrated significantly better scalability in terms of cost function over unseen scenarios with larger task/robot spaces than those used for training. Lastly, evidence regarding the unique potential of learning-based approaches in delivering highly time-efficient solutions is provided for a benchmark vehicle routing problem – where solutions are achieved 100-1000 times faster compared to a non-learning baseline, and for a benchmark MRTA problem with time and capacity constraints – where solutions for larger problems are achieved 10 times faster compared to non-learning baselines.

## 1 INTRODUCTION

In multi-robot task allocation (MRTA) problems, we study how to coordinate tasks among a team of cooperative robotic systems such that the decisions are free of conflict and optimize a quantity of interest (Gerkey & Matarić, 2004). The potential real-world applications of MRTA are immense, considering that multi-robotics is one of the most important emerging directions of robotics research and development, and task allocation is fundamental to most multi-robotic or swarm-robotic operations. Example applications include disaster response (Ghassemi & Chowdhury, 2018), last-mile delivery (Aurambout et al., 2019), environment monitoring (Espina et al., 2011), reconnaissance (Olson et al., 2012) and combat Behjat et al. (2021). Although various approaches (e.g., graph-based methods (Ghassemi et al., 2019), integer-linear programming (ILP) approaches (Nallusamy et al., 2009; Toth & Vigo, 2014), and auction-based methods (Dias et al., 2006; Schneider et al., 2015)) have been proposed to solve the ***combinatorial optimization*** problem underlying MRTA operations, they usually do not scale well with number of robots and/or tasks, and do not readily adapt to complex problem characteristics without tedious hand-crafting of the underlying heuristics. In the recent years, a rich body of work has emerged on using learning-based techniques to model solutions or intelligent heuristics for combinatorial optimization (CO) problems over graphs. The existing methods are mostly limited to classical CO problems, such as multi-traveling salesman (mTSP), vehicle routing (VRP), and max-cut type of problems. We specifically focus on a class of MRTA problems that falls into the Single-task Robots, and Single-robot Tasks (SR-ST) class defined in (Gerkey & Matarić, 2004; Nunes et al., 2017). Based on iTax taxonomoy as defined in Gerkey & Matarić (2004), these problems fall into the In-schedule Dependencies (ID) category. Here, a feasible and conflict-free task

allocation is defined as assigning any task to only one robot (Ghassemi et al., 2019). For solving these problems, we propose a new **covariant attention-based model** (aka **CAM**), a neural architecture for learning over graphs to construct the MRTA policies. This architecture builds upon the attention mechanism concept and innovatively integrates an equivariant embedding of the graph to capture graph structure while remaining agnostic to node ordering. We implement CAM on an original MRTA problem, a suite of benchmark MRTA problems and benchmark VRP problems to perform generalizability, scalability and comparative analyses of the new method. We also perform an analysis on the impact of the neighborhood size on the performance on the benchmark MRTA problem.

## 1.1 MULTI-ROBOT TASK ALLOCATION

In recent years, learning approaches based on Graph Neural Networks or GNN are being increasingly used to solve planning problems with a CO formulation, e.g., TSP, VRP, Max-Cut, Min-Vertex, and MRTA Kool et al. (2019); Barrett et al. (2019); Khalil et al. (2017); Kaempfer & Wolf (2018); Mittal et al. (2019); Li et al. (2018); Nowak et al. (2017); Wang & Gombolay (2020); Tolstaya et al. (2020); Sykora et al. (2020); Dai et al. (2017). Further details on these related learning-based studies can be found in Appendix A. Some of the conventional ILP, MILP, and INLP based methods for MRTA have been discussed in Appendix D.1. GNNs provide the advantage of being able to capture both Euclidean features (e.g., task location), as well as non-Euclidean features such as task capacity, task deadline and the local structure of task neighborhoods. The latter serves as higher-level meaningful features that assist in generalized decision-making These existing studies are however limited in three key aspects: **1)** They address simplified problems that often exclude common real-world factors such as resource and capacity constraints Kool et al. (2019); Kaempfer & Wolf (2018); Khalil et al. (2017); Tolstaya et al. (2020)). **2)** They are mostly focused on smaller sized problems ($\leq 100$ tasks and 10 robots) Paleja et al. (2020); Strens & Windelinckx (2005); Wang & Gombolay (2020); Sykora et al. (2020), with their scalability remaining unclear. **3)** They rarely provide evidence of generalizing to problem scenarios that are larger in size than those used for training. This capability would be particularly critical since real-world MRTA problems often involve simulating episodes whose costs scale with the number of tasks and robots, making re-training efforts burdensome. To address these gaps, here we propose a new learning framework that can solve large-sized MRTA problems (SR-ST) with commonly considered constraints – involving up to 1000+ tasks and 200+ robots – and generalize across even larger problem scenarios without the need to re-train. For most practical scenarios with larger number of locations, a highly optimal solution is not always desired, while a good feasible solution is the priority as pointed out by Cappart et al. (2021). Therefore, to enable scalable policies, we design a novel encoder based on the concept of Covariant Compositional Networks (CCN) Hy et al. (2018), which is hypothesized to effectively combine local structural information with permutation invariance. The encoder is followed by a decoder based on a Multi-head Attention mechanism Kool et al. (2019); Vaswani et al. (2017) which fuses the encoded information and problem/mission-specific information (Context) using simple matrix multiplication, in order to enable decentralized sequential decision-making.

## 1.2 CONTRIBUTIONS OF THIS PAPER

The primary contributions of this paper can thus be stated as follows: **1)** We formulate the general SR-ST class of MRTA problems as a Markov Decision Process or MDP over graphs with the multi-robot state information embedded as the *context* portion of the policy model, such that the (task allocation) policy can be learned using an RL approach. **2)** We design the GNN that acts as the policy network as an *encoder-decoder* architecture, where the *encoder* is innovatively based on covariant compositional networks (CCN), whose embedding capabilities significantly improve generalizability and scalability to larger task graphs and multi-robot teams. **3)** We implement an attention based decoder (inspired by Kool et al. (2019)) to enable sequential decision-making, and specifically extend it to a multi-agent combinatorial optimization setting. The proposed CAM architecture is evaluated on a representative MRTA problem that involves coordinating a team of unmanned aerial vehicles (UAVs) to time-efficiently deliver flood relief. The results of this case study demonstrate how CAM clearly outperforms the state-of-the-art attention based method AM (Kool et al., 2019), in terms of scalabilty and convergence, thereby emphasizing the effectiveness of the new encoder. Further case studies show that CAM continues to compare favorably to AM over benchmark MRTA problems (with time and capacity constraints) and CVRP problems. Comparisons to non-learning baselines for these benchmark problems demonstrate the significant online computation advantages of learnt policies, with the latter being 10-100 times faster. The remainder of the paper is organized as follows: Section 2 defines the MRTA problem and its formulation as an MDP over graphs. Section 3 presents

our proposed new GNN architecture. Section 4 describes simulation settings and different case studies. Results are discussed in Section 5.

## 2 MRTA: PROBLEM DEFINITION AND FORMULATIONS

The multi-robot task allocation (MRTA) problem is defined as the allocation of tasks and resources among several robots that act together without conflict in the same environment to accomplish a common mission. The optimum solution (decision) of the MRTA problem is a sequence of tasks for each robot (conflict-free allocation) that maximizes the mission outcome (e.g., fraction of tasks completed) or minimize the mission cost (e.g., total distance travelled) subject to the robots' range constraints. Here, the following assumptions are made: 1) All robots are identical and start/end at the same depot; 2) There are no environmental uncertainties; 3) The location $(x_i, y_i)$ of task-$i$ and its time deadline $\tau_i$ are known to all robots; 4) Each robot can share its state and its world view with other robots; and 5) There is a depot (Task-0), where each robot starts from and visits if no other tasks are feasible to be undertaken due to the lack of available range. Each tour is defined as departing from the depot, undertaking at least one task, and returning to the depot. 6) Motivated by the multi-UAV relief delivery problem, tasks are considered to be instantaneous, which means that reaching the waypoint associated with a task completes that task. This MRTA problem is a class of combinatorial optimization problems, which can be modeled in graph space. In order to learn policies that yield solutions to this CO problem, we express the MRTA problem as a Markov Decision Process (MDP) over a graph, described next. The optimization formulation of MRTA is then given in Section 2.2.

### 2.1 MDP OVER A GRAPH

The MRTA problem involves a set of nodes/vertices ($V$) and a set of edges ($E$) that connect the vertices to each other, which can be represented as a complete graph $\mathcal{G} = (V, E)$. Each node represents a task, and each edge connects a pair of nodes. Let $\Omega$ be a weight matrix where the weight of the edge ($\omega_{ij} \in \Omega$) represents the cost (e.g., distance) incurred by a robot to take task-$j$ after achieving task-$i$. For MRTA with $N$ tasks, the number of vertices and the number of edges are $N$ and $N(N-1)/2$, respectively. Node $i$ is assigned a 3-dimensional feature vector denoting the task location and time deadline, i.e., $d_i = [x_i, y_i, \tau_i]$ where $i \in [1, N]$. Here, $\omega_{ij}$ can be computed as $\omega_{ij} = \sqrt{(x_i - x_j)^2 + (y_i - y_j)^2}$, where $i, j \in [1, N]$.

The MDP defined in a decentralized manner for each individual robot (to capture its task selection process) can be expressed as a tuple $< \mathcal{S}, \mathcal{A}, \mathcal{P}_a, \mathcal{R} >$. The components of the MDP can be defined as: **State Space ($\mathcal{S}$):** A robot at its decision-making instance uses a state $s \in \mathcal{S}$, which contains the following information: 1) the current mission time, 2) its current location, 3) its remaining ferry-range (battery state), 4) the planned (allocated) tasks of its peers, 5) the remaining ferry-range of its peers, and 6) the states of tasks. The states of tasks contain the location, the time deadline, and the task status – active, completed, and missed (i.e., deadline is passed). Here we assume that each robot can broadcast its information to its peers without the need for a centralized system for communication, as aligned with modern communication capabilities Sykora et al. (2020). **Action Space ($\mathcal{A}$):** The set of actions is represented as $\mathcal{A}$, where each action $a$ is defined as the index of the selected task, $\{0, \ldots, N\}$ with the index of the depot as 0. The task 0 (the depot) can be selected by multiple robots, but the other tasks are allowed to be chosen once if they are active (not completed or missed tasks). $\mathcal{P}_a(s'|s, a)$**:** A robot by taking action $a$ at state $s$ reaches the next state $s'$ in a deterministic manner (i.e., deterministic transition model is defined). **Reward ($\mathcal{R}$):** The reward function is defined as $-f_{\text{cost}}$, and is calculated when there is no more active tasks (all tasks has been visited once irrespective of it being completed or missed). **Transition:** The transition is an event-based trigger. An event is defined as the condition that a robot reaches its selected task or visits the depot location.

### 2.2 MRTA AS OPTIMIZATION PROBLEM

This MRTA problem is adopted from (Ghassemi et al., 2019; Ghassemi & Chowdhury, 2021) with the following modification – payload constraints are not imposed on the robot. The exact solution of the MRTA problem can be obtained by formulating it as an integer nonlinear programming problem, which can be summarily expressed as:

$$\min \ f_{\text{cost}} = r - u(r)e^{-d_r} \tag{1}$$

$$\text{where } r \in [0, 1] \ \text{ and } \ u(r) = \begin{cases} 1 & \text{if } r = 0 \\ 0 & \text{otherwise} \end{cases} \tag{2}$$

$$\text{subject to} \quad \tau_i^{\text{f}} < \tau_i \tag{3}$$

$$\delta_{ij} \le \Delta_k, k \in [1, N_r] \ i, j \in [0, N] \tag{4}$$

Here $\tau_i^{f}$ is the time at which task $i$ was completed, $\Delta_k$ is the available range for robot $k$ at any point of time, and $\delta_{ij}$ is the distance between nodes $i$ and $j$. A detailed formulation of the exact ILP constraints that describe the MRTA problem with range restrictions, multi-tours per robot and tasks with deadlines, can be found (Ghassemi & Chowdhury, 2021). Note that, here we use a slightly different objective/cost function to better reflect the generalized setting for the class of MRTA problems with ferry-range and task-deadline constraints For compactness of representation, only the main constraints involved in the studied MRTA problem are shown in the above set of equations. We however consider all of the constraints, except the one related to payload capacity, as defined in the work by (Ghassemi & Chowdhury, 2021); for detailed formulation of the MRTA problem, please refer to (Ghassemi & Chowdhury, 2021).

Here, we craft the objective function (Eq. equation 1) such that it emphasises maximizing the completion rate (i.e., the number of completed tasks divided by the total number of tasks); and if perfect completion rate (100%) is feasible, then the travelled cost is also considered. The term of $1 - r$ is defined as task completion rate; i.e., the number of completed tasks ($N_{\text{success}}$) divided by the total tasks ($N$) or $r = (N - N_{\text{success}})/N$. Here, $d_r$ is a normalized value of the total distance travelled by all robots in the team. The term $d_r$ is the average travelled distance over all robots (i.e., $d_r = \sum_{i=1}^{N_r} d_i^{\text{total}}/(\sqrt{2}\,N)$). The terms $N_r$ and $d_i^{\text{total}}$ represent respectively the number of robots and the total traveled distance by robot-$i$ during the entire mission. The above objective function (Eq. equation 1) gives a positive value if the completion rate is lower than 100%, otherwise it gives a negative value. Equation 1 ensures that the objective function is bounded in the range $(-1, 1]$.

## 3 COVARIANT ATTENTION-BASED NEURAL ARCHITECTURE

For learning to work on the MDP defined over graphs in Section 2.1, we need to represent each node as a continuous vector, preserving its properties as well as the the structural information of the neighborhood of that node.

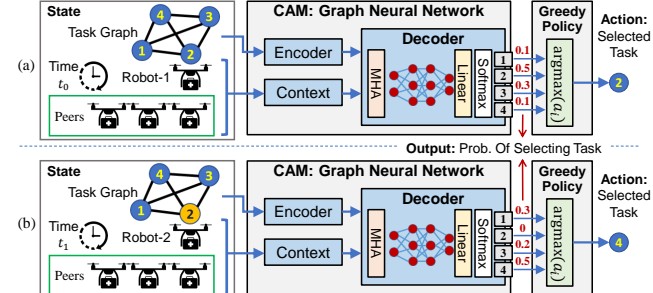

Before describing the technical components of our proposed Covariant Attention Mechanism, the so-called CAM neural architecture, we provide an illustration and summary description here of how this policy architecture is used by robots or agents during an SR-ST operation. The CAM model for task allocation is called by/for each robot right when it reaches its current destination (task location or depot), in order to decide its next task or destination. While robots are moving toward their selected task locations, they can check if their decision is conflicting with another robot based on recent information. If there is conflict, the robot with the worst time can cancel its current task and select a new task. Since full-observability is assumed across the multi-robot team and the policy-model execution-time is almost negligible, the current setup is agnostic to whether the online CAM model is executed centrally off-board or on-board each robot. As an example, Figure 1 illustrates how robot-1 and robot-2 uses the CAM policy model to choose a task at two different decision-making instances ($t = t_0$ and $t = t_1$). Here, the inputs to the CAM model includes **1)** the task graph information, i.e., the properties of all the tasks/nodes $d_i$ and the computed weight matrix $\Omega$, **2)** the current mission time, **3)** the state of robot-$r$, and **4)** the states of robot-$r$'s peers. The CAM model then generates the probability of selecting each task as its output. A greedy strategy of choosing the task with the highest probability is used here, which thus provides the next destination to be visited by that robot. It should be noted that the probability values for completed tasks and missed tasks (i.e., missed deadline) are set at 0.

Figure 1: Deployment of an MRTA policy using CAM architecture. a) Robot-1 at $t_0$. b) Robot-2 at $t_1$; here, the CAM output for previously selected task (task 2 in (b)) is set at 0.

Figure 2 shows the detailed architecture of CAM. As shown in this figure, the CAM model consists of three key components, namely: ***Context***, ***Encoder***, and ***Decoder***. The context includes the current mission time, the states of robot-$r$, and the states of robot-$r$'s peers. The state of a robot consists of it's destination $x, y$ coordinates and the available range $\rho$. The encoder and decoder components are described below.

## 3.1 CCN-INSPIRED NODE ENCODER

For learning over graphs, the performance of the trained model depends mostly on the ability of the Graph Neural Network (GNN) to transform all the required node information into a feature vector or tensor. For our case, apart from the node properties, some of the features that is essential include a node's local neighborhood information, and permutation invariance. Using a node's local neighborhood information which consists of its association with its local neighbors during training is

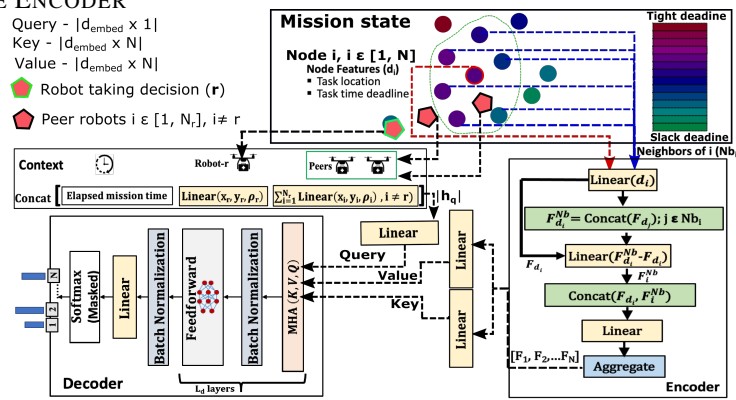

Figure 2: CAM architecture with the information flow along the context, encoder & decoder. The node are represented by the colored circles, where the colors represents the time deadline. The robots are represented by the pentagon shapes and the robot highlighted (green) is the robot taking decision.

more beneficial than considering the association with the entire graph, for generalizing to unseen nodes as demonstrated by frameworks like GraphSAGE (Hamilton et al., 2017), thus enabling the GNN to generalize for problems with larger number of nodes without the need to re-train. The encoder represent the properties of each graph node (preserving its structural information) into a continuous feature vector of dimension $d_{embed}$, which is fed to the decoder. Each node $i$, has three properties which are the x-coordinate ($x$), y-coordinate ($y$), and the time deadline ($\tau$) of the task. Note that our encoding mechanism can also be extended to a probabilistic scenario, for example where an estimated deadline $\tau$ follows a probability distribution (common in disaster response type operations). The encoding for each node should include its properties and the its positional association with its neighboring nodes. We implement a variation of CCN (Hy et al., 2018). We determine the nearest $k$ neighbors of a node ($Nb_i$) based on the positional coordinates ($x$ and $y$). The first step is to compute a feature vector by linear transformation for each node $i$. To encode the node properties, we do a linear transformation of $d_i$ to get a feature vector $F_{d_i}$ for all $i \in [1, N]$, i.e., $F_{d_i} = W^d d_i^T + b_d$.

Here $W^d \, \varepsilon \, \mathbb{R}^{d_{embed} \times 3}$, $b_d \, \varepsilon \, \mathbb{R}^{d_{embed} \times 1}$, $d_i = [x_i, y_i, \tau_i]$. For effective decision making, we also need to preserve the structural information. Therefore we define a matrix $F_{d_i}^{Nb}$ as in equation 5.

$$F_{d_i}^{Nb} = \text{Concat}(F_{d_j}), \quad j \in Nb_i \tag{5}$$

We compute a matrix $F_i^{Nb}$ (as shown in Eq. equation 6), which we believe captures the association of a node with its local neighbours (one-hop neighbors) in terms of the node properties.

$$F_i^{Nb} = W^{Nb}(F_{d_i}^{Nb} - F_{d_i}) + b^{Nb} \tag{6}$$

where $W^{Nb} \, \varepsilon \, \mathbb{R}^{d_{embed} \times d_{embed}}$, $b^{Nb} \, \varepsilon \, \mathbb{R}^{d_{embed} \times 1}$. $F_i^{Nb}$ captures the information about how close the node properties of neighbor nodes of node $i$ is to itself, which shows a representation of how important node $i$ is to its neighbors. Further explanation regarding this design choice is discussed in Appendix B and we strongly encourage the reader to go over the section.

We compute the final embedding for each node using Eq. equation 7.

$$F_i = \text{Aggregate}(W_f(\text{Concat}(F_i^d, F_i^{Nb})) + b_f) \tag{7}$$

Here, $W_f \, \varepsilon \, \mathbb{R}^{d_{embed} \times d_{embed}}$, $b_f \, \varepsilon \, \mathbb{R}^{d_{embed} \times 1}$. Thus finally we get an embedding $F_i$ for each node, where $F_i \, \varepsilon \, \mathbb{R}^{d_{embed} \times 1}$. $W^d$, $b_d$, $W^{Nb}$, $b^{Nb}$, $W_f$, and $b_f$ are learnable weights and biases. The Aggregate function is the summation across all the columns of a matrix. This summation along with the relative difference in node properties, as in Eq. equation 6, preserves permutation-invariance and the structural properties (cognizance of inter-node distances for example) of the graph. Note that, these operations make the encoded state w.r.t. a given node insensitive to the order of the neighboring nodes, and thus the overall state space becomes independent of the indexing of tasks or to graph rotations. Equations 5, 6, and 7 represents a single layer of encoding. Multiple layers of encoding can be performed with the output of the previous layer being the inputs to equations 5 and 6 in the next layer.

## 3.2 ATTENTION-BASED DECODER AND CONTEXT

The main objective of the decoder is to use the information from the encoder and the current state as context or query, and thereof choose the best task by calculating the probability value of getting selected for each (task) node. In this case, the first step is to feed the embedding for each node (from the encoder) as **key-values** ($\mathcal{K}, \mathcal{V}$). The key $\mathcal{K}$ and value $\mathcal{V}$ for each node is computed by two separate linear transformations of the node embedding obtained from the encoder. The next step is to compute a vector representing the current state, also known as the **context** (as shown in bottom left of Fig. 2). The context for the MHA layer in this experiment consist of the following seven features: 1) Current time; 2) Available range of the robot taking decision; 3) Current location of robot taking decision; 4) Current destination of other robots; 5) Available range for other robots all concatenated to a single vector of length $h_q$, which then undergoes a linear transformation to get a vector of length $d_{embed}$ also called the query $Q$. Figure 2 illustrates the structure of the decoder.

Now the attention mechanism can be described as mapping the query ($\mathcal{Q}$) to a set of key-value ($\mathcal{K}, \mathcal{V}$) pairs. The inputs, which are the query ($\mathcal{Q}$) is a vector, while $\mathcal{K}$ and $\mathcal{V}$ are matrices of size $d_{embed} \times N$ (since there are $N$ nodes). The output is a weighted sum of the values $\mathcal{V}$, with the weight vector computed using the compatibility function expressed as:

$$\text{Attention}(\mathcal{K}, \mathcal{V}, \mathcal{Q}) = \text{softmax}(Q^T \mathcal{K}/\sqrt{d_{embed}})\mathcal{V}^T \tag{8}$$

Here $h_l$ is the dimension of the key of any node $i$ ($k_i \in \mathcal{K}$). In this work, we implement a multi-head attention (MHA) layer in order to determine the compatibility of $\mathcal{Q}$ with $\mathcal{K}$ and $\mathcal{V}$. The MHA implemented in this work is similar to the decoder implemented in Kool et al. (2019) and Vaswani et al. (2017). As shown in (Vaswani et al., 2017) the MHA layer can be defined as:

$$\text{MHA}(\mathcal{K}, \mathcal{V}, \mathcal{Q}) = \text{Linear}(\text{Concat}(\text{head}_1 \dots \text{head}_{h_e})) \tag{9}$$

Here $\text{head}_i = \text{Attention}(\mathcal{K}, \mathcal{V}, \mathcal{Q})$ and $h_e$ (taken as 8 here) is the number of heads. The feed-forward layer is implemented to further process the mapping that results from the MHA layer, and transform it to a dimension that is coherent with the number of nodes in the task-graph ($N$). The interjecting batch normalization layers serve to bound values of a specific batch using the mean and variance of the batch. The final `softmax` layer outputs the probability values for all the nodes. Here, the next task to be done is then chosen based on a greedy approach, which means that the node with the highest probability will be chosen. The nodes which are already visited will be masked (by setting its probability as 0) so that these nodes are not available for selection in the future time steps of the simulation of the multi-robot operation.

## 4 CASE STUDIES

We design and execute a set of numerical experiments described in Appendix D.3, to investigate the performance of our proposed learning-based algorithm over graph space (*CAM*) and compare it with 1) an extended version of a state-of-the-art graph learning-based algorithm proposed by (Kool et al., 2019), so called *attention-based mechanism* (*AM*) approach; 2) a recent bipartite graph matching method BiG-MRTA (Ghassemi & Chowdhury, 2021); 3) a myopic baseline called Feasibility-preserving Random-Walk (Feas-RND) that takes randomized but feasible actions (avoiding conflicts and satisfying other problem constraints). Since the BiG-MRTA method has been shown to provide near-optimal solutions in comparison to ILP (and competitive performance w.r.t. state-of-the-art online MRTA methods) Ghassemi & Chowdhury (2021), it is used here to comparatively gauge performance of AM and CAM. The Feas-RND method on the other hand provides a baseline that AM and CAM should clearly surpass in performance (cost function), in order to demonstrate that meaningful MRTA policies are being learnt as opposed to simply mapping random feasible actions.

## 5 RESULTS AND DISCUSSION

### 5.1 GENERALIZABILITY AND SCALABILITY ANALYSIS OF CAM

The CAM model has been trained on scenarios with 200 tasks and varying robot size (randomly a robot size between 10 and 50 has been selected). Then, 100 test scenarios have been generated per robot-task size from the same distribution of training scenarios. In this paper, *generalizability* refers to the performance of the trained model on unseen test scenarios that involve the same (or lower) number of tasks as in the scenarios used for training; and where the test and training scenarios are drawn from the same probability distribution over task locations and deadlines. In this work, generalizability was analysed on test scenarios with the number of tasks fixed at 50 and 200, drawn from the same distribution over a 2D space, and number of robots fixed at 5 and 40. In this paper, *scalability* refers to the performance of the trained model over test scenarios with higher (and increasing) numbers of tasks and robots than that encountered in the scenarios used for training. Here, we analyze scalability

by evaluating the CAM model on test scenarios with the number of tasks fixed at 500 and 1000, and the number of robots fixed at 50 and 1000. The task-to-robot ratio is however kept the same across the generalizability and scalability analysis cases, in order not to introduce another control factor affecting the numerical experiments. To measure and compare performance, we use two metrics: 1) Average cost function (Eq. 1): This metric accounts for the completion rate of tasks and the total travelled distance, averaged over the set of test scenarios; and 2) Average computing time: This measures how long each method takes to compute the entire solution, averaged over the test set. The latter is particularly important to note in scalability analysis, since state-of-the-art non-learning based MRTA methods scale poorly in terms of computing efficiency as numbers of tasks and robots increases.

Table 1 shows the cost function (the lower the better) for the unseen test scenarios for both AM and CAM. Note that the cost function (as defined by Eq.1) for MRTA produces a negative value when the completion rate is

Table 1: **MRTA – Multi-UAV flood response:** Generalizability: Task size (up to 200) and number of robots (up to 40). Average cost function (with the average computation time in seconds). Lower the better.

| # of Tasks | # of Robots | Avg. Cost Function (Avg. Computing Time in seconds) | | | |
|---|---|---|---|---|---|
| | | **BiG-MRTA** | **Feas-RND** | **AM** | **CAM** |
| 50 | 5 | **-0.27** (0.21) | 0.27 (0.5) | 0.07 (0.14) | -0.03 (0.14) |
| | 10 | **-0.70** (0.20) | 0.03 (0.3) | -0.20 (0.15) | -0.47 (0.16) |
| 100 | 10 | **-0.47** (0.80) | 0. 31 (1.2) | 0.10 (0.33) | 0.01 (0.32) |
| | 20 | **-0.72** (1.21) | 0.05 (0.8) | -0.16 (0.34) | -0.50 (0.34) |
| 200 | 20 | **-0.66** (4.72) | 0.34 (1.9) | 0.09 (1.34) | 0.04 (1.21) |
| | 40 | **-0.74** (8.37) | 0.05 (1.1) | -0.07 (1.43) | -0.45 (1.40) |

100%, while a positive value indicates a task completion rate below 100%. As it can be seen from Table. 1 that the proposed CAM approach outperforms the AM approach in all the test cases by achieving better mean values of the cost function, respectively. The CAM approach performs significantly better than AM in terms of the cost function for the lower task-to-robot ratio (here, 5). Based on table 1, the proposed CAM approach achieved a perfect completion rate for most of scenarios with the task-to-robot ratio of 5. The local structure of the graph is not only important for effective decision-making, but also expected to be shared across various problems settings drawn from the same distribution, thereby promoting generalizability of policies when adequately captured.

To investigate the scalability of the learnt model, we use a new set of unseen test scenarios with numbers of tasks and robots much larger than those of the scenarios used in training (the latter involved 200 tasks).

Table 2: **MRTA – Multi-UAV flood response:** Scalability w.r.t. number of tasks and robots; showing average cost function (and avg. computation time in sec). Lower the better.

| # of Tasks | # of Robots | Avg. Cost Function (Avg. Computing Time in seconds) | | | |
|---|---|---|---|---|---|
| | | **BiG-MRTA** | **Feas-RND** | **AM** | **CAM** |
| 500 | 50 | **-0.73** (69.3) | 0.34 (4.2) | 0.10 (4.23) | 0.07 (4.31) |
| | 100 | **-0.77** (135.3) | 0.05 (2.6) | -0.02 (4.29) | -0.37 (4.71) |
| 1000 | 100 | **-0.76** (595.4) | 0.34 (8.7) | 0.09 (19.22) | 0.06 (19.05) |
| | 200 | **-0.76** (1420) | 0.06 (5.4) | 0.02 (20.00) | -0.31 (20.03) |

Table 2 shows the performance of the trained model of CAM and AM in terms of the cost function (lower the better) for four large case studies, involving 500-tasks-50-robots, 500-tasks-100-robots, 1000-tasks-100-robots, and 1000-tasks-200-robots; for each case, 100 randomly generated scenarios (i.e., task locations and deadlines randomized) are used. As shown in Table 2, the proposed CAM method outperforms the AM method in all cases, with a significant difference in the cost function for the case studies with the task-to-robot ratio of 5. In the largest case (i.e., 1000-task-200-robot), the learnt model by CAM achieved a perfect completion rate for most of the scenarios (Appendix E). It can be argued that, for time-critical problems such as MRTA in disaster response, generating an optimal solution is less of a priority, compared to generating a feasible near-optimal solution as quickly as possible; this capability of the learnt models in evident from the results in Table 7 in Appendix E. BiG-MRTA requires a significantly larger computing time for larger sized problem (see Fig. 5 in Appendix E.1). Comparison with the BiG-MRTA solutions also indicate remaining scope of improvement for learning methods, in terms of distance travelled.

The comparison with Feas-RND shows how the learning method is markedly better than random **feasible** myopic decisions, thereby indicating that meaningful MRTA policies have been learnt here, as opposed to producing random feasible solution by virtue of the masked policy network design. In the Feas-RND method, a robot takes a task-selection decision randomly from a feasible set of choices that abide by all the constraints in the problems, e.g., related to inter-robot conflicts, task deadline and robot range. As shown in Table 7, CAM achieves more than 92% task completion rate for all the scenarios, which is found to be a 5%-30% gain in task completion rate over that achieved by

Feas-RND across these scenarios. The performance of CAM in terms of the task completion rate is comparable to that of BiG-MRTA (which is slightly better), with the highest difference being 6.9% for the 500-tasks-50-robots case. Note that CAM continues to be better than AM is all these cases as well. Later in Section 5.2, we also compare the performance of CAM, AM, and BiG-MRTA on a benchmark MRTA problem where the objective is just to maximize the task completion rate (hence the cost function is not affected by distance travelled), where CAM proves to be highly competitive.

**Ablation study:** We performed two ablation studies (Appendix E.2) on CAM to understand the importance of the novel encoder and the decoder (adopted from attention mechanism). In the first ablation study, the CCN based encoding is replaced with simple feedforward layers (as explained in Appendix E.2), with the decoder remaining the same. It should be noted that the node embedding length ($d_{embed}$), is the same for all the cases. In the second ablation study, the MHA based decoding was replaced with simple feedforward layers and a softmax layer (as explained in Appendix E.2), with the encoder remaining the same. As shown by Table 8, in both of the cases (i.e., with the encoder and decoder respectively ablated), we observe a significant decrease in the performance across all scenarios, with the maximum dip in the completion rate being 19.8% and 13.4% for the first and the second study, respectively. Compiling the results from the first ablation study for the encoder and the comparison of CAM with AM, we posit that the CCN based encoding, which is able to aggregate local node neighborhoods while remaining agnostic to node ordering, clearly aids in providing better policies. Similarly, from the second case of the ablation study, we can conclude the MHA based decoding, which computes a compatibility of the current state information with the node information, aid in learning better policies.

5.2 COMPARATIVE ANALYSIS ON BENCHMARK MRTA PROBLEMS AND CVRP PROBLEMS

***MRTA - Task Allocation Problem with Time and Capacity (TAPTC)***: The CAM architecture is implemented and tested on a well-known class of (NP hard) MRTA problems known as Task Allocation Problem with Time and Capacity constraints or TAPTC, as described in Mitiche et al. (2019). In TAPTC, each task $i$ has a time deadline ($t_i$) and workload ($w_i$), and each robot has a work capacity ($c_j$). The time to finish task $i$ by robot $j$ is defined as $w_i/c_j$. We compare the results of CAM on TAPTC with that of AM (Kool et al., 2019) and with those of the three non-learning baseline methods, namely 1) Iterated Local Search (ILS) (Vansteenwegen et al., 2009), which uses a meta heuristic approach, 2) Enhanced Iterated Local Search (EILS) (Vansteenwegen et al., 2009), which has controlled runtime and perturbations (compared to ILS), and 3) Bi-Graph MRTA (BiG-MRTA) (Ghassemi & Chowdhury, 2021)), which uses a bigraph construction and maximum weighted matching approach. Further details of the TAPTC benchmark, the baselines used here, and the changes made to CAM and AM for this case study are discussed in Appendix F. For the results shown here, the CAM model is implemented such that $k = 9$ nearest neighbors are considered when computing the embedding of each node of the task graph. The testing is performed for different scenarios that are characterized by the number of robots and % of tasks having slack time deadline. Mitiche et al. (2019) categorized the TAPTC problems into 2 groups based on the value of the high deadline for the tasks. Table 3 here presents the results for group 2, with the group 1 results given in Table 10 in Appendix F.4. In these tables, the scenario nomenclature is defined as: R75A5 denotes $R = 75\%$ tasks have a normally-distributed time deadline and a team of $A = 5$ robots. Results on the impact of neighborhood size ($k$) on CAM performance is also discussed in Appendix F.4. Table 3 shows that for all scenarios with $A = \{5, 7\}$, CAM performs better than both AM and the non-learning baselines in terms of task completion rate; and CAM achieves top performance for 50% of the scenarios with $A = \{2, 3\}$.

Table 3: **MRTA - TAPTC** Group 2: Performance of CAM, AM and baselines in terms of average completion rate. Bold style indicate the best value for the scenario. Higher the better.

| | Avg. Completion Rate | | | | | | | | | | | | | | | |
| | R25A2 | R25A3 | R25A5 | R25A7 | R50A2 | R50A3 | R50A5 | R50A7 | R75A2 | R75A3 | R75A5 | R75A7 | R100A2 | R100A3 | R100A5 | R100A7 |
|---|---|---|---|---|---|---|---|---|---|---|---|---|---|---|---|---|
| ILS | 45.00 | 63.33 | 95.67 | **100.00** | 42.33 | 61.67 | 91.67 | **100.00** | 40.33 | 58.33 | 83.67 | **100.00** | 33.33 | 50.33 | 76.33 | 96.33 |
| EILS | 45.67 | 64.00 | 98.00 | **100.00** | 44.00 | 63.33 | 94.33 | **100.00** | 41.00 | **59.33** | 86.33 | **100.00** | **35.67** | **50.67** | 78.00 | 98.67 |
| BiG-MRTA | 46.00 | 65.67 | 90.67 | 98.33 | 45.33 | 63.67 | 87.33 | 96.67 | **42.00** | 57.00 | 79.00 | 95.00 | 34.33 | 50.00 | 73.00 | 91.67 |
| AM | 47.33 | 65.33 | **100.00** | **100.00** | 29.00 | 64.67 | **100.00** | **100.00** | 23.67 | 39.67 | **100.00** | **100.00** | 20.33 | 36.33 | 76.67 | **100.00** |
| CAM(k=9) | **53.00** | **78.67** | **100.00** | **100.00** | **45.67** | **74.33** | **100.00** | **100.00** | 31.00 | 55.00 | **100.00** | **100.00** | 29.33 | 45.33 | **99.67** | **100.00** |

Table 4: **MRTA - TAPTC** Group 2: Average computation time (in milliseconds) to generate the entire solution for each scenario.

| | Avg. Computation Time | | | | | | | | | | | | | | | |
| | R25A2 | R25A3 | R25A5 | R25A7 | R50A2 | R50A3 | R50A5 | R50A7 | R75A2 | R75A3 | R75A5 | R75A7 | R100A2 | R100A3 | R100A5 | R100A7 |
|---|---|---|---|---|---|---|---|---|---|---|---|---|---|---|---|---|
| EILS | 1233.33 | 678.33 | 1041.67 | 14.67 | 920.00 | 796.67 | 1010.33 | 23.33 | 1008.67 | 1714.67 | 2541.67 | 598.33 | 168.33 | 1791.00 | 1416.00 | 1005.33 |
| BiG-MRTA | 147.71 | 226.07 | 392.70 | 716.49 | 108.55 | 211.31 | 409.40 | 624.28 | 109.76 | 259.32 | 479.83 | 753.97 | 160.10 | 325.24 | 581.50 | 686.86 |
| AM | 96.60 | 93.48 | 94.15 | 94.31 | 93.33 | 93.67 | 94.14 | 94.15 | 94.39 | 102.23 | 93.97 | 93.46 | 95.59 | 114.82 | 106.12 | 105.07 |
| CAM(k=9) | 114.40 | 93.01 | 94.31 | 94.57 | 94.79 | 94.87 | 96.10 | 94.39 | 94.54 | 105.57 | 95.90 | 94.31 | 94.84 | 94.33 | 95.55 | 95.74 |

A comparison of the computation time to generate the entire MRTA solution, shown in Table 4, demonstrates the advantage of learning based methods over non-learning baselines as the problem size increases w.r.t. the number of robots and tasks. ***Capacitated VRP***: To demonstrate the versatility of the proposed CAM architecture, we train and test the CAM and the AM architectures (for comparison) on a benchmark variation of the Vehicle Routing Problem (VRP), known as the Capacitated VRP (CVRP). The CVRP benchmark consists of $N$ task locations, where a vehicle is required to visit the locations and deliver packages in a manner that minimizes a cost function. We also use the Lin-Kernighan heuristics (LKH3) solver (Helsgaun, 2017) and the Simulated Annealing (SA) algorithm implementation provided by Google Operations Research (OR) tools as well-regarded non-learning based baselines for comparing the results obtained by CAM and AM. Further details regarding the CVRP problem, the baseline method, and settings changes for CAM and AM are presented in Appendix G.1. Table 5 summarizes the results of all four approaches (i.e., AM, CAM, and Google OR) on unseen scenarios for varying task size (# of locations) ranging from 50 to 1,000, in terms of the cumulative computing time and the cost function (both as average values across scenarios of a given size). LKH3 is a well known state-of-the-art method for solving CVRP problems and also has the best performance (considering only cost and not run time). As shown in Table 5, the performance of all methods except LKH3 are comparable for small sized problems (50, 100, and 200 locations or tasks). As expected, the main advantage of the proposed CAM approach is apparent for the problems with larger number of tasks (i.e., 500 and 1000 tasks), where the average cost function of the solutions obtained by the CAM is significantly better (less than half of AM's and one-third of Google OR's). In these larger-sized scenarios, the computing time performance of CAM is slightly better than that of AM, and together AM and CAM are two orders of magnitude faster than SA (Google OR) and LKH3 in generating the entire solution (sequence of tasks to be undertaken).

## 6 CONCLUSION

Table 5: **CVRP**: Comparison of average cost function (and average time taken to generate the entire solution). Lower the better.

| # OF TASKS | AVG. COST FUNCTION (AVG. COMPUTING TIME) | | | |
|---|---|---|---|---|
| | **LKH3** | **GOOGLE OR** | **AM** | **CAM** |
| 50 | **10.53** ($46s$) | 11.3 ($2s$) | 12.3 ($0.04s$) | 12.2 ($0.04s$) |
| 100 | **15.58** ($60s$) | 17.6 ($5s$) | 17.4 ($0.09s$) | 17.9 ($0.09s$) |
| 200 | **17.69** ($86s$) | 21.3 ($20s$) | 21.6 ($0.18s$) | 21.8 ($0.17s$) |
| 500 | **24.87** ($123s$) | 54.5 ($20s$) | 34.0 ($0.53s$) | 29.1 ($0.53s$) |
| 1000 | **28.67** ($189s$) | 81.8 (200$s$) | 64.1 ($1.51s$) | 41.6 ($1.49s$) |

In this paper, we proposed a new GNN architecture, called CAM, for a multi-robot task allocation problem with a set of complexities, including tasks with time deadline and robots with constrained range. This new architecture incorporates an encoder based on covariant node-based embedding and a decoder based on attention mechanism. A simple RL algorithm has been implemented for learning the features of the encoder and decoder. In addition, to compare the performance of the proposed CAM method, an attention-based mechanism approach (aka AM) has been extended to be able to handle a multi-agent CO setting problem, along with a recent state-of-the-art method BiG-MRTA, and a mypoic baseline method Feas-RND. To evaluate the performance of the proposed CAM architecture and the extended version of AM, they are trained with the same settings. All the methods were tested on 100 unseen case studies. Performance was analyzed in terms of the cost value and the completion rate. Our primary proposition, CAM, outperformed AM and Feas-RND on test scenarios by achieving better cost function value, and was also able to achieve high task completion rate ($> 92\%$) for even larger sized problems without the need to retrain, which is comparable to the near-optimal (but $O(10^1 - 10^2)$ more expensively computed) solutions by BiG-MRTA, thereby demonstrating the favorable scalability of CAM. The computational cost analysis showed that the proposed CAM model takes a few milliseconds to compute a decision, thereby providing clear advantage over non-learning based approaches to MRTA in the context of online (time-sensitive) planning. Moreover, the advantage of using local neighborhood information for node encoding can be seen in the scalability analysis on the MRTA and CVRP, where CAM demonstrates superior performance when applied to graphs with larger number of tasks/nodes. The ablation studies in Appendix E.2 showed the importance of the CCN based encoding and the MHA based decoding used in CAM. A comparisons over a different suite of benchmark MRTA problems showed that CAM was competitive w.r.t. standard non-learning baselines including BiG-MRTA, in terms of task completion rate, while providing substantially faster solutions compared to the latter. Lastly, a comparative analysis on different CAM models with varying neighborhood size (while encoding) was performed (appendix F.6) to study the impact of the neighborhood size $k$ in the encoder. Based on Appendix F.6 (for MRTA-TAPTC), k is a trade off between accuracy and computation time.

**Ethics statement:** The work in this paper does not have any direct negative societal consequences.

**Reproducibility statement:** The CAM architecture which includes the encoder, decoder and the context part, can be coded in any programming language by following the equations in section 3 and figure 2. The dataset for training the MRTA-Multi-UAV Flood response problem can be generated using the information in Appendix D.3. The codes for the AM method can be obtained from `https://github.com/wouterkool/attention-learn-to-route`. The AM method can be modified for solving the MRTA-Multi-UAV Flood Response problem using the information from Appendix D.5. Both CAM and AM method can be trained using the settings in table 6. The codes for the BiG-MRTA method and Feas-RND can be obtained from `https://github.com/adamslab-ub/BiG-MRTA`. The training data for MRTA-TAPTC can be generated using the information in Appendix F.5. Both CAM and AM can be modified using the information in Appendix F.2. Details on running BiG-MRTA for MRTA- TAPTC can be obtained from Ghassemi et al. (2019) and the corresponding code from `https://github.com/adamslab-ub/BiG-MRTA`. The EILS can be coded using the inforation in Mitiche et al. (2019). The test dataset can be obtained from `https://tinyurl.com/taptc15in`. The CAM architecture can be modified for CVRP using the information in Appendix G.1.1 and G.1.2. The codes for the AM method and implementation for LKH3 can be obtained from `https://github.com/wouterkool/attention-learn-to-route`. The google OR tools implementation can be done using the example in `https://developers.google.com/optimization/routing/vrp` with the dataset generated using the information in Appendix G.4.

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

# A  LEARNING OVER GRAPHS

Neural network based methods for learning CO can be broadly classified into: (i) Reinforcement Learning (RL) methods Kool et al. (2019); Barrett et al. (2019); Khalil et al. (2017); Strens & Windelinckx (2005); and (ii) Supervised learning (often combined with RL) methods Kaempfer & Wolf (2018); Mittal et al. (2019); Li et al. (2018); Nowak et al. (2017). The supervised learning approaches typically address problem scenarios where samples are abundant (e.g., influence maximization in social networks Mittal et al. (2019)) or inexpensive to evaluate (e.g., TSP Kaempfer & Wolf (2018)), and are thus unlikely to be readily applicable to solve complex problems over real-world graphs. RL based techniques to learn on graphs include attention models with REINFORCE Kool et al. (2019) and deep Q-learning Khalil et al. (2017); Barrett et al. (2019), among others, with some extending solutions to multi-agent settings Jiang et al. (2020). In this work, we are interested in the first class of the methods (i.e., RL methods over graph space).

Dai et al. Dai et al. (2017) showed that a combination of graph embedding and RL methods can be used to approximate optimal solutions for combinatorial optimization problems, as long as the training and test samples are drawn from the same distribution. Mittal et al. (2019) presented a new framework to solve a combinatorial optimization problem. In this framework, Graph Convolutional Network (GCN) performs the graph embedding and Q-Learning learns the policy. The results demonstrated that the proposed framework can learn to solve unseen test problems that have been drawn from the same distribution as that of the training data set. More importantly, it has been shown that using a learned network policy instead of tree search, both methods are using the same embedding GCN, showed a speedup of 5.5 for a problem size of 20,000 for influence maximization. Similarly, the effectiveness of learning a network policy using Q-Learning to solve the Max Cut problem (a combinatorial problem) has been demonstrated by Barrett et al. (2019).

Recently, there has been a growing interest in using sequence-to-sequence models (e.g., pointer networks and attention mechanism) to encode and learn the combinatorial optimization problems in graph space Kaempfer & Wolf (2018); Kool et al. (2019). Kool et al. Kool et al. (2019) implemented a framework using an encoder/decoder based on attention mechanism and REINFORCE algorithm for solving a wide variety of combinatorial optimization problems as graphs, with the main contribution being flexibility of the approach on multiple problems with the same hyperparameters. Wang et al. Wang & Gombolay (2020) showed how learning can lead to generate faster solutions than standard exact methods for multi-robot scheduling problems. However, the size of the problem that has been studied in that work and other similar studies Paleja et al. (2020); Sykora et al. (2020) is limited to 5 robots and 100 tasks, and only temporal constraints were considered. In this paper, we study larger problems (up to 1000 tasks/200 robots) as well as include complexities such as time deadlines for tasks, robot ferry range constraints, capacity constraints, multiple routes, etc. Graph learning has been implemented for a multi-robot coverage problem in Tolstaya et al. (2020), which demonstrates good scalability. However, this work is addressing a multi-robot exploration problem (and not MRTA), and it is not clear how this proposed method can be applied to an MRTA problem with complexities such as range constrain, capacity constraints, time extended tasks, multiple routes, etc. Apart from CO problems, graph learning can also be used to perform path planning as demonstrated in Li et al. (2020).

In this paper, a new neural architecture is proposed that combines the *attention mechanism* with an enhanced encoding network (embedding layers), where the latter is particularly designed to capture local structural features of the graph in an equivariant manner. The embedding layer is a variation of *Covariant Compositional Networks* (CCN), introduced by Hy et al. (2018). CCN was originally implemented for predicting molecular properties by learning local structural information of the molecules. This node-based embedding has been chosen since it: i) operates on an undirected graph; ii) uses receptive field and aggregation functions based on tensor product and contraction operations (which are equivariant), which leads to a permutation- and rotation-invariant embedding; and iii) provides an extendable representation ($n$-th order tensor representation can be useful to extend the work to multi-level networks, e.g., involving multiple node properties). We found an exact implementation of the CCN to be computationally burdensome for learning policies in large MRTA problems, and hence a variation of the CCN is proposed in this work. An attention mechanism has been successfully implemented for problems with sequential processes, e.g., Natural Language Processing (NLP) Vaswani et al. (2017). Here we are interested in the attention mechanisms since they involve simple matrix multiplications, which make them not only computationally inexpensive

(by utilizing modern GPUs), but also programmatically easy to be implemented. In this work, we implement an attention-based decoder for CAM as proposed in Kool et al. (2019); Vaswani et al. (2017).

## B  EMBEDDING LOCAL STRUCTURAL INFORMATION

Local structural or neighborhood information of a graph node/ task refers to the association of the node to its neighboring nodes. These information includes how the properties of the neighborhood nodes differ from that of the node under consideration. It is this information that is being encoded in the node embedding along with the information of the node properties $d_i$. The encoder should be able to use this local structural information in order to scale to larger-sized problems as pointed out by (Cappart et al., 2021). Here, we explain how local structural information is being encoded with the help of two graphs, one which is small and the other is a larger graph but having some nodes with similar neighborhood as shown in figure 3. Node $A$ in graph $G1$, and nodes $B$, and $C$ in graph $G2$ have almost similar neighborhood. Here the local neighborhood information encoding of nodes $A$, $B$, and $C$ (which are $F_A^{Nb}$, $F_B^{Nb}$, and $F_C^{Nb}$ respectively) will be almost same, irrespective of the size of the graph. Therefore by equation 6, $\left((F_{d_A}^{Nb} - F_{d_A}) \approx (F_{d_B}^{Nb} - F_{d_B}) \approx (F_{d_C}^{Nb} - F_{d_C})\right) \implies$ $(F_A^{Nb} \approx F_B^{Nb} \approx F_C^{Nb})$. Even though the actual locations of nodes $A$, $B$, and $C$ are different, their association with it's neighboring are almost the same, which is being captured by equation 6.

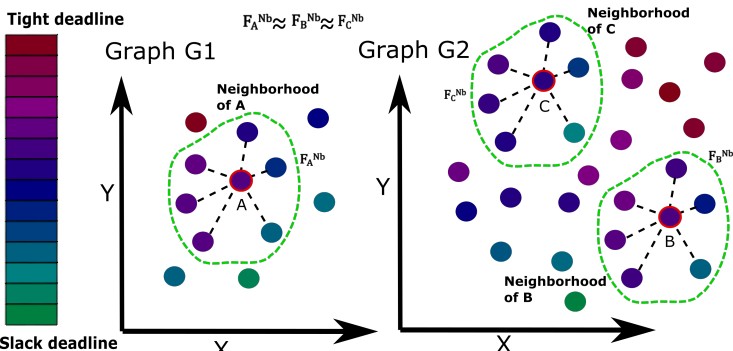

Figure 3: Node $A$ belongs to a smaller sized graph $\mathcal{G}1$, while $B$ and $C$ are nodes of a larger-sized graph $\mathcal{G}2$. Based on the relative positions of its neighbor nodes (represented by the dashed lines), and the deadline of neighbor nodes (color of the nodes), $A$, $B$, and $C$ has a similar neighborhood.

## C  LEARNING FRAMEWORK

Both the CCN-inspired encoder and the attention-based decoder consist of learnable weight matrices as explained in Sections 3.1 and 3.2. In order to learn these weight matrices, both supervised and unsupervised learning methods can be used. However, supervised learning methods are not tractable since the computational complexity of the exact I(N)LP solution process required to generate labels. The complexity of the ILP formulation of the MRTA problem scales with $O(n^3 m^2 h^2)$, where $n$, $m$, and $h$ represent the number of tasks, the number of robots, and the maximum number of tours per robot, respectively Therefore, we use a reinforcement learning algorithm to conduct the learning. **Learning Method:** In this work we implement a simple policy gradient method (REINFORCE) as the learning algorithm with greedy rollout baseline, which also enables us to compare the effectiveness of our method with that of (Kool et al., 2019). For each epoch, two sets of data are used which are the training set and the validation set. The training data set is used to train the training model ($\theta_{\text{CAM}}$) while the validation data set is used to update the baseline model ($\theta_{\text{CAM}}^{BL}$). The size of the training data and the validation data used for this paper is mentioned in section D.3. Each sample data from the training and validation data-set consist of a graph as defined in Section 2.1. The pseudo code of the training algorithm for our architecture is shown in Alg. 1 in Appendix E. It should be noted that the policy gradient method requires the evaluation of a cost function, which is defined to be same as in Eq. equation 1. **Policy:** We define the policy such that if the robot $r$ does not satisfy the constraints in Eqs. equation 3, it returns to depot (i.e., $a = 0$). Otherwise the robot $r$ runs the learnt CAM network

and chooses the output (task) based on a greedy approach (selects a task with the highest probability value), as shown in Fig. 1.

---

**Algorithm 1** Training Algorithm

---

**Input:** $N_E$: Number of epochs, $N_b$: Number of batch, $B$: Batch size, $N_{tr}$: Training data size, $N_{vl}$: Validation data size.

1: $\theta_{\text{CAM-RL}}$ - CAM-RL
2: $\theta_{\text{CAM-RL}}^{BL}$ - Baseline CAM-RL
3: **for** epoch = $1..N_{\text{epoch}}$ **do**
4:      $\mathcal{D}_{tr}, \mathcal{D}_{vl} \leftarrow$ GenerateScenarios($N_{tr}, N_{vl}$)
5:      $N_b \leftarrow \lfloor N_{tr}/B \rfloor$
6:      **for** step = $1..N_b$ **do**
7:          $\mathcal{D}_{tr,b} \leftarrow$ SampleRandom($\mathcal{D}_{tr}, M$) {$\mathcal{D}_{tr,b}$: Batch Training Dataset}
8:          $\mathbf{a}^{\text{BL}}, f_{\text{cost}}^{\text{BL}} \leftarrow$ CalculateCost($\theta_{\text{CAM-RL}}^{BL}, \mathcal{D}_{tr,b}$)
9:          $\mathbf{a}, f_{\text{cost}} \leftarrow$ CalculateCost($\theta_{\text{CAM-RL}}, \mathcal{D}_{tr,b}$)
10:         $\nabla\mathcal{L} \leftarrow \frac{1}{B}\sum_{i=1}^{B}(f_{\text{cost,i}} - f_{\text{cost,i}}^{\text{BL}})\log\text{softmax}(\mathbf{a}_i)$
11:         $\theta_{\text{CAM-RL}} \leftarrow$ ADAM($\nabla\mathcal{L}, \theta_{\text{CAM-RL}}$)
12:      **end for**
13:      $\mathbf{a}_{vl}^{\text{BL}}, f_{\text{cost,vl}}^{\text{BL}} \leftarrow$ CalculateCost($\theta_{\text{CAM-RL}}^{BL}, \mathcal{D}_{vl}$)
14:      $\mathbf{a}, f_{\text{cost}} \leftarrow$ CalculateCost($\theta_{\text{CAM-RL}}, \mathcal{D}_{vl}$)
15:      **if** ($\sum_{i=1}^{N_{vl}} f_{\text{cost,i}}^{\text{BL}} > \sum_{i=1}^{N_{vl}} f_{\text{cost,i}}$) $\wedge$ (T-Test($\mathbf{a}_{vl}, \mathbf{a}_{vl}^{\text{BL}}$) $> \epsilon$) **then**
16:         $\theta_{\text{CAM-RL}}^{BL} \leftarrow \theta_{\text{CAM-RL}}$
17:      **end if**
18: **end for**
19: **CalcuateCost Procedure:**
20: **for** $i = 1..|\mathcal{D}|$ **do**
21:      $\mathbf{a}_i, f_{\text{cost,i}} \leftarrow$ Simulation($\theta, \mathcal{D}_i$)
22:      $\mathbf{a} \leftarrow \mathbf{a} \cup \mathbf{a}_i$
23:      $f_{\text{cost}} \leftarrow f_{\text{cost}} \cup f_{\text{cost,i}}$
24: **end for**
25: **return** $\mathbf{a}, f_{\text{cost}}$

---

### C.1 SIMULATION AND FRAMEWORK SETTINGS

The *"Python"* 3.7 and the 64-bit distribution of *"Anaconda 2020.02"* are used to implement the MRTA approaches. The environment, training algorithm, and the evaluation of the trained model, are all implemented in *Pytorch-1.5* for CAM and AM. The training, based on Pytorch, is deployed on two GPUs (NVIDIA Tesla V100) with 16GB RAM.

## D MORE DETAILS ON MRTA

### D.1 CONVENTIONAL METHODS FOR MRTA

The MRTA problem can be formulated as an Integer Linear Programming (ILP), mixed ILP or Integer Non-Linear Programming (INLP) problem depending on the application. When tasks are defined in terms of location, the MRTA problem becomes analogical to the Multi-Traveling Salesmen Problem (mTSP) (Khamis et al., 2015) and its generalized version, the Vehicle Route Planning (VRP) problem (Dantzig & Ramser, 1959). Existing solutions to mTSP and VRP problems in the literature (Bektas, 2006; Braekers et al., 2016) have addressed analogical problem characteristics of interest to MRTA, albeit in a disparate manner; these characteristics include tasks with time deadlines, and multiple tours per vehicle, with applications in the operations research and logistics communities (Azi et al., 2010; Wang et al., 2018). ILP-based mTSP-type formulations and solution methods have also been extended to task allocation problems in the multi-robotic domain (Jose & Pratihar, 2016). Although the ILP-based approaches can in theory provide optimal solutions, due to the NP-hard time complexity of the SR-ST problems (Mazyavkina et al., 2021; Archetti et al., 2011), they are characterized by exploding computational effort as the number of robots and tasks increases (Toth & Vigo, 2014; Cattaruzza et al., 2016). For example, for the studied SR-ST problem, the cost of solving the exact integer programming formulation of the problem scales with $\mathcal{O}(n^3 m^2 h^2)$, where $n$, $m$, and $h$ represent the number of tasks, the number of robots, and the maximum number

of tours per robot, respectively (Ghassemi et al., 2019). As a result, most practical online MRTA methods, e.g., auction-based methods (Dias et al., 2006) and bi-graph matching methods (Ghassemi & Chowdhury, 2018; Ismail & Sun, 2017), use some sort of heuristics, and often report the optimality gap at least for smaller test cases compared to the exact ILP or INLP solutions (where tractable).

## D.2 MORE DETAILS ON ENCODING THE CONTEXT

As discussed in section 3.2, the context portion of while a robot $r$ makes a decision, consists of 1) Current time $t$; 2) Available range of the robot taking decision $\rho_r$; 3) Current location of robot taking decision $(x_r, y_r)$; 4) Current destination of other robots $(x_p, y_p, \forall\, p \in P_r)$; 5) Available range for other robots $(\rho_p, \forall\, p \in P_r)$, where $P_r$ represents the peers of robot $r$. Here, features 2 and 3 represent the current state of the robot taking the decision, while features 4 and 4 represents the states of the peer robots. The context feature vector can be computed as shown in Eq.10.

$$\mathcal{Q} = Linear(Concat(t, Q_r, Q_{P_r})) \tag{10}$$

where,

$$Q_r = Linear([x_r, y_r, \rho_r] \tag{11}$$

and

$$Q_{P_r} = \Sigma_{p \in P_r} Linear([x_p, y_p, \rho_p]) \tag{12}$$

The dimensions of $Q_r$, and $Q_{P_r}$ is $d_{embed}$, and the length of the final feature vector $Q$ is also considered $d_{embed}$. The summation aggregation operation in Eq.12, makes the context vector agnostic to the number of robots.

## D.3 DESIGN OF EXPERIMENTS & LEARNING PROCEDURES

To evaluate the proposed CAM method, we define an MRTA case study with varying number of UAVs and 200 task (flood victims) locations. A 2D environment with 1 sq. km area is used for this purpose, with the time deadline of tasks varied from 0.1 to 1 hour. The UAVs are assumed to have a range of 4 km, and a nominal speed of 10 km/h. We assume instantaneous battery swap is provided at depot location, which is used when UAVs return to depot since they were running low on battery. It is important to note that the flood victim application is used here merely for motivation, and the CAM architecture is in no way restricted to this application, but can rather solve problems in the broad (important) class of capacity/range-constrained and timed task-constrained SR-ST problems. Moreover, even the policies learnt here for CAM demonstration on the described case settings can generalize to related SR-ST problems with up to 1,000 tasks, which represents a fairly large MRTA problem in reference to the existing literature in the MR domain.

To perform learning and testing of the learned model, we proceed as follows: *Learning Phase:* We use a policy gradient reinforcement learning algorithm (REINFORCE with rollout baselines in this case) for learning the optimal policy. The learnable parameters in this architecture includes all the weights in the encoder and the decoder. The training is carried out for a total of 100 epochs. Each epoch consists of 10,000 random training samples, which are evaluated and trained in batches of 100 samples. *Testing Phase:* In order to provide a statistically insightful evaluation and comparison, the models are tested for different cases of varying number of tasks and varying number of robots with each case having 100 random test scenarios from training data distribution. Here, for each task in a sample scenario, the location of the tasks, time deadline, location of the depot are all generated from a random uniform distribution. More details on the learning framework is given in Appendix C. The training and testing settings and the modifications to AM for MRTA are given in Appendix D.

## D.4 BASELINES

**BiG-MRTA:** The BiG-MRTA algorithm Ghassemi & Chowdhury (2021); Ghassemi et al. (2019) is an online method based on the construction and maximum weighted matching of a bipartite graph. BiG-MRTA (Ghassemi & Chowdhury, 2021) uses a novel combination of a bipartite graph construction, an incentive model to assign edge weights in the bigraph, and maximum weighted matching (based on the Karp algorithm (Karp, 1980)) over the bigraph to allocate tasks to robots. This method has been developed as an online solver for SR-ST type MRTA problems, where tasks have

deadlines, new tasks could appear during the mission, and robots are subject to range and payload constraints.

**AM**: The number of attention heads for the encoder is 8, with 3 layers of encoding. The node embedding length is 128.

**Feas-RND:** In the Feas-RND approach, each robot randomly chooses available tasks that are feasible to be undertaken by the UAV, satisfying all the constraints in section 2.2. The algorithm used for Feas-RND can be found in (Ghassemi & Chowdhury, 2021).

### D.5    MORE DETAILS ON TRAINING FOR MRTA

**Learning Curve** : In order to compare the convergence of the proposed CAM method with that of the AM approach, we run both methods with similar settings and plot their learning curve (convergence history), as shown in Fig. 4 in appendix D. As seen from this figure, the AM method took 3 epochs to reach its best cost value and stagnated. On the other hand, the CAM method took 20 epochs to reach the best cost value of AM and continued to improve up to $\sim$24 epochs, leading to a significantly better cost function value ($f^*_{\text{cost,CAM}} = -0.266$) compared to AM ($f^*_{\text{cost,AM}} = -0.009$). The stagnation of AM could be attributed to direct implementation of the transformer network (Vaswani et al., 2017), which was designed for machine translation and thus consists of multiple layers of Multi-head attention. In contrast, our CAM model uses simple linear transformations of the node properties and its relative differences in local neighborhoods to capture structural information.

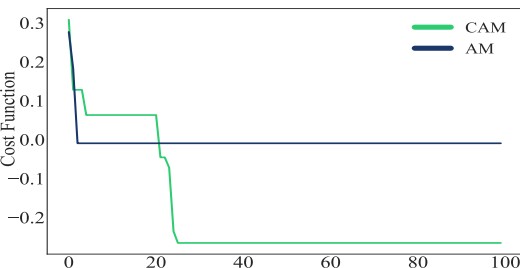

Figure 4: **MRTA:** Learning curve of CAM and AM for 200 tasks.

**Modifications to AM**: The attention-based mechanism (AM) reported by (Kool et al., 2019) has been shown to solve a few different classes of single-agent/robot combinatorial optimization problems. To be able to implement the AM method for our problem (for comparison with our CAM method), the AM method is adapted to a multi-robot setting. For this purpose, we make the following three changes to the AM method: (i) The node properties that are defined in Section 2.1 are used in AM; (ii) The context for the attention mechanism is modified to be the same as that used for CAM; and (iii) The cost function used for training is changed to that in Eq. equation 1.

We want to compare the structural representational quality of CAM and AM for generalizability and scalabilty. Hence, for a fair comparison, both CAM and AM models were trained with the same settings (Table 6 and14).

#### D.5.1    COMPUTING TIME (TRAINING AND EXECUTION)

Based on the epoch information in Section D.3, the average time to complete a training epoch was found to be 19.50 minutes (i.e., $\sim$11.7 seconds per sample) for CAM and AM. The average computing time (from Tables 1 and 2) taken by the learnt policies for producing the entire MRTA solution (sequence of tasks assigned to each robot) was found to grow from  0.14 s to  20 s, as the number of tasks grew from 50 to 1000, and thus remaining unprecedentedly attractive for real-time decision-making even for large problems.

## E    MRTA – MULTI-UAV FLOOD RESPONSE: FURTHER RESULTS

### E.1    TASK COMPLETION RATE AND COMPUTATION TIME

Table 7 shows the completion rate which corresponds to the test cases for generalizability (Table 1) and the test for scalability (Table.2), respectively.

Table 6: **MRTA**: Training algorithm settings for CAM and AM for MRTA-TAPTC

| DETAILS | AM | CAM |
|---|---|---|
| *Algorithm* | *REINFORCE* | *REINFORCE* |
| *Baseline* | *Rollout* | *Rollout* |
| *Epochs* | 100 | 100 |
| *# of tasks* | 200 | 200 |
| *Training samples* | 10000 | 10000 |
| *Baseline samples* | 1000 | 1000 |
| *Optimizer* | *Adam* | *Adam* |
| *Learning step size* | 0.0001 | 0.0001 |
| *Training frequency* | 100 SAMPLES | 100 SAMPLES |

Table 7: **MRTA – Multi-UAV flood response:** Comparison of CAM and AM on completion rate. Higher the better.

| # of Tasks | # of Robots | Avg. Completion Rate | | | |
|---|---|---|---|---|---|
| | | **BiG-MRTA** | **Feas-RND** | **AM** | **CAM** |
| 50 | 5 | 96.82 | 63.85 | 89.8 | 92.8 |
| | 10 | 99.88 | 93.54 | 98.0 | 99.2 |
| 100 | 10 | 99.15 | 64.45 | 91.8 | 92.5 |
| | 20 | 99.98 | 94.04 | 98.0 | 99.8 |
| 200 | 20 | 99.91 | 65.06 | 93.85 | 95.05 |
| | 40 | 100.00 | 94.59 | 98.65 | 100.0 |
| 500 | 50 | 99.93 | 65.18 | 92.64 | 93.00 |
| | 100 | 100.00 | 94.16 | 98.42 | 99.92 |
| 1000 | 100 | 99.95 | 65.33 | 93.26 | 93.77 |
| | 200 | 100.00 | 94.22 | 98.45 | 99.95 |

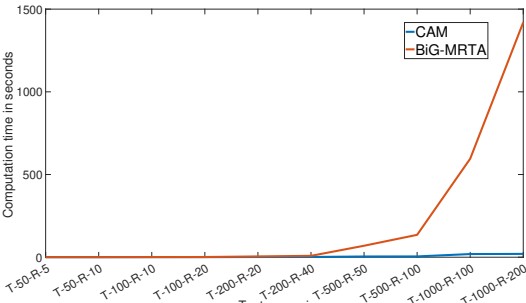

Figure 5: **MRTA-Multi-UAV flood response:** Comparison of the average computation time in seconds for CAM and BiG-MRTA, for the different scenarios. The x-axis denotes the different scenarios. For example, T-50-R-5 corresponds to the scenarios with 50 tasks and 5 robots.

Figure 5, shows the comparison of the average computation time between CAM and BiG-MRTA. It can be seen that, the computation time of BiG-MRTA increases exponentially as compared to that of CAM. Even though the completion rate for BiG-MRTA is slightly greater than that of CAM as can be seen from table 7, this advantage comes with a very high computational cost. This shows the scalability of CAM to larger-sized problems. On comparing the cost function in tables 1 and 2, it can be seen that, compared to BiG-MRTA, the performance of CAM drops for scenarios with higher tasks/robot ratios (such as 200-tasks-20-robots, 500-tasks-50-robots). However, this dip in performance is marginal when comparing the task completion rate (in tables 7), with the maximum dip in performance for CAM being 7.3% (between 100-tasks-10-robots and 100-tasks-20-robots). One of the reason for this behaviour, is due to the cost function (Eq.1), where the distance minimization is only taken into account when the task completion rate is 100%. Hence, we also compared the performance of CAM, AM, and BiG-MRTA on a benchmark MRTA problem in section 5.2, where

the objective is just to maximize the task completion rate, in which CAM demonstrates a superior performance compared to the other methods.

## E.2   ABLATION STUDIES

Table 8: **MRTA – Multi-UAV flood response: Ablation studies**. The performance is compared with respect to the average cost function of 100 testing scenarios (with % task completion rate). $\mathbf{CAM}_{E_{FF}}$: CAM model with CCN-based encoding ablated; $\mathbf{CAM}_{D_{FF}}$ CAM model with AM based decoding ablated

| # of # tasks | # of # robots | Avg. Cost Function (Avg. Completion Rate) | | |
|---|---|---|---|---|
| | | **CAM** | $\mathbf{CAM}_{E_{FF}}$ | $\mathbf{CAM}_{D_{FF}}$ |
| 50 | 5 | -0.03 (92.8) | 0.26 (73.0) | 0.20 (79.8) |
| | 10 | -0.47 (99.2) | -0.07 (95.6) | 0.07 (92.4) |
| 100 | 10 | 0.01 (92.5) | 0.26 (73.6) | 0.19 (80.8) |
| | 20 | -0.50 (99.8) | -0.01 (96.0) | 0.05 (93.3) |
| 200 | 20 | 0.04 (95.0) | 0.24 (75.5) | 0.19 (81.6) |
| | 40 | -0.45 (100.0) | 0.04 (95.5) | 0.06 (93.9) |
| 500 | 50 | 0.07 (93.0) | 0.23 (76.3) | 0.19 (80.4) |
| | 100 | -0.37 (99.9) | 0.04 (95.1) | 0.07 (92.9) |
| 1000 | 100 | 0.06 (93.7) | 0.12 (77.9) | 0.18 (81.0) |
| | 200 | -0.31 (99.9) | 0.02 (95.0) | 0.05 (94.0) |

**For Encoder:**  In order to study the true impact of the graph node encoding, we compared the performance of CAM with it's covariant based encoding removed to get a new model $\mathbf{CAM}_{E_{FF}}$. In this case the, node encoding is performed using a simple feedforward network, following Eq.14. The decoder for $CAM_{E_{FF}}$ is the same as that of $CAM$.

$$F_{d_i} = W^d d_i^T + b_d \tag{13}$$

$$F_i = W^F F_{d_i} + b_F \tag{14}$$

where, $d_i = [x_i, y_i, \tau_i], \forall i \in V$. $W^d, W^F, b_d$, and $b_F$ are learnable weights and biases, where $W^d \varepsilon \mathbb{R}^{d_{\text{embed}} \times 3}$, $b_d \varepsilon \mathbb{R}^{d_{\text{embed}} \times 1}$, and $W^F \varepsilon \mathbb{R}^{d_{\text{embed}} \times d_{\text{embed}}}$, $b_F \varepsilon \mathbb{R}^{d_{\text{embed}} \times 1}$

**For Decoder:**  In order to study the impact of the MHA based decoder, we compare the performance of CAM, with it's decoder being replaced by a simple feedforward network to get a new model $\mathbf{CAM}_{D_{FF}}$, which takes in the node embeddings and the context information, and computes the output probabilities using the following equations. The encoder for $CAM_{D_{FF}}$ is the same as that of $CAM$.

$$P^{Act} = softmax([P_1^{Act}, \dots, P_N^{Act}]), \tag{15}$$

$$P_i^{Act} = \text{LeakyReLU}(W^{dec}Concat(F_i, \mathcal{Q})^T + b_{dec}), \ where \ i \ \in \ V \tag{16}$$

Here $W^{dec}$ and $b_{dec}$ are learnable weights and biases, where $W^{dec} \in \mathbb{R}^{N \times 2d_{\text{embed}}}$ and $b_{dec} \in \mathbb{R}^{N \times 1}$. $F_i \ \forall \ i \in V$ are the node embeddings from the encoder, and $\mathcal{Q} \in \mathbb{R}^{d_{\text{embed}}}$ is the context vector. The value of $d_{\text{embed}}$ for both the models is the same as that of CAM in section 5. Both the models were trained on the MRTA-Multi-UAV flood response problem in section 4, using the parameters in table 6, and tested on the same test scenarios as that used for $CAM$, $AM$, and $BiG\text{-}MRTA$, for varying number of locations and varying number of robots (shown in table 8). The third and the fourth column in table 8 shows the average cost function of 100 test scenarios, and it's corresponding % completion rate, for $CAM_{E_{FF}}$ and $CAM_{D_{FF}}$ ablation study, respectively.

As can be seen from table 8, the performance of both $CAM_{E_{FF}}$ and $CAM_{D_{FF}}$ is significantly poor compared to $CAM$, in terms of both cost function and task completion rate. The performance drop is more for scenarios with lower number of robots, with the maximum drop in task completion rate being for $CAM_{E_{FF}}$ being 19.8% (for 50-tasks-5-robots), and for $CAM_{D_{FF}}$ being 13.4% (for 200-tasks-20-robots). It can also be observed that, the performance drop is greater for $CAM_{E_{FF}}$ than $CAM_{D_{FF}}$, this indicates that the CCN-based encoder has a slightly more influence on the performance, than the decoder.

# F    MRTA - TASK ALLOCATION PROBLEM W/ TIME & CAPACITY (TAPTC)

## F.1    PROBLEM DESCRIPTION AND FORMULATION

In order to assess the effectiveness of the CAM architecture for MRTA problems with time-extended assignments, we test and validate our CAM on Task Allocation Problem with Time and Capacity (TAPTC) benchmark problem Mitiche et al. (2019), which falls into the ST-SR-TA-TW (Single Task robot, Single Robot task, Time-extended Assignment, TimeWindows). The TAPTC involves task locations within a $100 \times 100$ grid map, with each task $i$ having a time deadline $t_i$, and also a workload $w_i$. Each robot $j$ has a work capacity of $c_j$. All the robots move with a maximum speed of 1 unit per second. A task $i$ is considered to be completed only if a robot $j$ visits node $i$ and spends a time of $w_i/c_j$. Each scenario is generated based on a uniform distribution reported in Mitiche et al. (2019) and the training samples for our models are generated from the same distribution.

For evaluating the performance of the learned model and conducting a comparative analysis, we use the TAPTC dataset (accessible from `http://tinyurl.com/taptc15in`. This dataset consists of test cases with 100 tasks and varying number of robots ($A = 2, 3, 5, 7$ robots), and the speed of every robot is considered to be the same (1 unit per second). The test cases can be divided equally into two groups, based on tight deadlines (Group 1) and slack deadlines (Group 2). Each group can be further divided into 4 sub-categories based on the fraction of tasks ($R = 25\%, 50\%, 75\%$, and $100\%$) that have normally distributed task deadlines. For example, $25\%$ indicates that there are 25 tasks with deadlines normally distributed between the limits $t_{\text{low}}$ and $t_{\text{high}}$, while the remaining 75 tasks have a deadline of $t_{\text{high}}$. For group 1 (tight deadline), the value of $t_{\text{high}}$ is considered as half as that for group 2 (tight deadline). In both the groups, for all values of $A$ and $R$, there will be 3 samples. Therefore the total number of test cases is given by `number of groups` $\times |A| \times |R| \times$ 3) = 96, where each group has 48 test cases. Further description of the test cases can be found in Mitiche et al. (2019).

The exact solution of the MRTA-TAPTC problem can be obtained by formulating it as the following integer non-linear programming (INLP) problem:

$$\min \ f_{\text{cost}} = \sum_{i=1}^{N} r_i, \quad \begin{cases} r_i = \frac{\tau_i^f}{\tau_i}, & \text{if } \tau_i^f > \tau_i \\ 0, & \text{otherwise} \end{cases} \tag{17}$$

subject to

$$s_i \in S \ \forall \ i \in [1..N] \tag{18}$$
$$s_i \neq s_j \ \forall \ i \neq j \tag{19}$$

Here, $N$ is the number of tasks/nodes, $\tau_i^f$ is the time at which task $i$ was completed, $\tau_i$ is the time deadline of task $i$, and $S = [s_1, s_2, ...s_N]$ is the sequence of all the nodes that were visited. The minimum cost function that can be achieved using Eq. 1 is 0, which corresponds to the case where all the tasks are successfully completed. A detailed formulation of the exact ILP constraints that describe the MRTA-TAPTC problem can be found in Mitiche et al. (2019). Note that, in our paper we use a slightly different reward function as compared to the objective function in Mitiche et al. (2019), but the intention of both the functions are essentially the same, which is to maximize the number of successfully completed tasks.

Here, we craft the objective function (Eq. equation 17) such that only missed tasks ($\tau_i^f > \tau_i$) contribute to the cost function. It is important to note that the objective function can be tailored according to the priority of the problem. Since the main priority in Mitiche et al. (2019) is to maximize the number of successfully completed tasks, the objective function (Eq. 17) also prioritizes task completion for a fair comparison with the baseline methods. The constraints in Eq. 18 and 19 are such that each tasks must be visited exactly once by any robot.

## F.2    MODIFICATIONS TO CAM AND AM

**1. Change in encoder:** The encoder for MRTA-TAPTC problems consider additional node properties, namely the location of the tasks $(x_i, y_i)$, the time deadline $(\tau_i)$, and the workload $w_i$, i.e., $d_i = [x_i, y_i, \tau_i, w_i]$.

**2. Change in context:** The context for the MHA layer in the decoder consists of the following five features: 1) elapsed mission time; 2) Work capacity of the robot taking decision; 3) Current location of robot taking decision; 4) Current destination of peers; and 5) Work capacity of peer.

**3. Cost function:** We use Eq. 17 as cost function.

The decoder need no change in this case.

### F.3  BASELINE METHODS

We consider three non-learning methods, which are i) Iterated Local Search (ILS), ii) Enhanced Iterated Local Search (EILS), iii) Bi-Graph MRTA (BiG-MRTA), which are briefly described below. The learning based baseline method we implemented here is AM with minor modifications as also discussed below.

**i) Iterated Local Search (ILS):** This is an online meta heuristic iterated search algorithm Vansteenwegen et al. (2009), where the output of one iteration is partially used as the input to the next iteration. During each iteration, the best solution is improved by a perturbation step, followed by a local search.

**ii) Enhanced Iterated Local Search (EILS):** EILS is also an online meta heuristic iterated search method Mitiche et al. (2019), with an improved perturbation step as compared to Vansteenwegen et al. (2009).

**iii) Bi-Graph MRTA (BiG-MRTA):** The BiG-MRTA algorithm Ghassemi & Chowdhury (2021); Ghassemi et al. (2019) is an online method based on the construction and maximum weighted matching of a bipartite graph. BiG-MRTA (Ghassemi & Chowdhury, 2021) uses a novel combination of a bipartite graph construction, an incentive model to assign edge weights in the bigraph, and maximum weighted matching (based on the Karp alogirthm (Karp, 1980)) over the bigraph to allocate tasks to robots. This method has been developed as an online solver for SR-ST type MRTA problems, where tasks have deadlines, new tasks could appear during the mission, and robots are subject to range and payload constraints.

**iii) AM:** The AM implementation is almost the same for the MRTA-TAPTC problem as that implemented for the multi-UAV flood response problem in section D.3. In addition, the changes implemented in CAM, as explained in Section F.2, are also applied here. The number of attention heads for the encoder is 8, with 3 layers of encoding. The node embedding length is 128.

Both CAM and AM are trained using REINFORCE as described in algorithm 1 using the settings given in Table 9.

Table 9: **MRTA - TAPTC:** Settings for model training for all CAM models and AM

| DETAILS | VALUES |
|---|---|
| *Algorithm* | *REINFORCE* |
| *Baseline* | *Rollout* |
| *Epochs* | 100 |
| *# of tasks* | 100 |
| *Training samples* | 500,000 |
| *Baseline samples* | 10,000 |
| *Optimizer* | *Adam* |
| *Learning step size* | 0.0001 |
| *Training frequency* | 500 SAMPLES |

### F.4  RESULT AND DISCUSSIONS

Tables 3 and 10 summarize the performance of CAM alongside the baseline methods, in terms of the average completion rate, i.e, the ratio of the number of successfully completed tasks to the total number of tasks averaged over 3 samples for all the scenarios (denoted by $A$ and $R$). The CAM model here uses $k = 9$, where $k$ represents the number of nearest neighbors considered for a node for computing its node embedding. For group 1, the CAM model was able to generate the best results for 8 out of the 16 different scenarios (Table 10). From Table 10, it can be inferred that CAM has a superior performance compared to the baselines for cases with larger number of robots (best performance for all cases with $A = 7$, and for 3 out of 4 scenarios with $A = 5$), including a maximum

margin of 19% for `R75A7` compared to the next best solution (EILS). While for cases with smaller number of robots ($A = \{2, 3\}$), CAM achieved the best performance for one scenario (`R25A3`), with the worst performance having a margin of only 7% (scenario `R75A3`) compared to best performer (EILS) in that case.

For group 2, CAM achieves top performance for all scenarios with $A = \{5, 7\}$, while achieving top performance for 50% of the scenarios with $A = 2, 3$, as discussed in Section 5.2 of the main text.

Tables 11 and 4 gives the average computation time (in milliseconds) to generate the entire solution for all the methods. As it can be seen from these tables, for the non-learning methods EILS and BiG-MRTA, the general trend is an increase in the computation time with increasing number of robots, while for both the learning-based methods (CAM and AM) the computation time increases marginally with increasing number of robots.

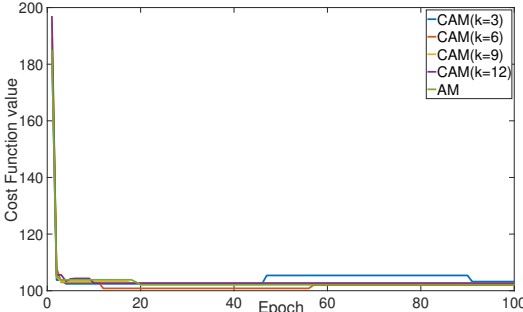

Figure 6: **MRTA-TAPTC:** Learning curve of all CAM models and AM model.

### F.5 MORE DETAILS ON TRAINING DATASET FOR MRTA-TAPTC

Each training sample has 100 tasks, and are located randomly within a $100 \times 100$ grid map. Each task $i$ has a time deadline $50 \leq \tau_i \leq 600$, and a workload $10 \leq w_i \leq 30$. Each sample has $n_r$ number of robots where $2 \leq n_r \leq 7$. The initial positions of the robots in a sample is also chosen randomly within the grids. Each robot $j$ has a work capacity of $c_j$ where $1 \leq c_j \leq 3$. All the robots move at a speed of 1 unit. A task $i$ is considered to be completed only if a robot $j$ visits node $i$ and spends a time of $w_i/c_j$. All the training samples are generated such that all the associated variables (mentioned above) follow a uniform distribution within their respective bounds.

Table 10: **MRTA - TAPTC** Group 1: Performance of CAM, AM and baselines in terms of average completion rate. Bold font values indicate the best performer for the corresponding scenario. Higher the better.

| | R25A2 | R25A3 | R25A5 | R25A7 | R50A2 | R50A3 | R50A5 | R50A7 | R75A2 | R75A3 | R75A5 | R75A7 | R100A2 | R100A3 | R100A5 | R100A7 |
|---|---|---|---|---|---|---|---|---|---|---|---|---|---|---|---|---|
| | | | | | | | Avg. Completion Rate | | | | | | | | | |
| ILS | 23.00 | 32.33 | 52.33 | 76.67 | 21.67 | 30.67 | 52.33 | 73.67 | 21.00 | 30.67 | 50.33 | 72.33 | 19.33 | 26.33 | 42.33 | 64.33 |
| EILS | 23.00 | 33.33 | 53.67 | 77.67 | 22.00 | **32.00** | 52.67 | 74.00 | **21.33** | **32.33** | **52.00** | 73.00 | 20.33 | **27.33** | 43.67 | 66.00 |
| BiG-MRTA | **23.33** | 33.00 | 51.33 | 77.33 | **23.33** | 32.00 | 53.33 | 70.67 | 21.00 | 31.67 | 49.00 | 71.00 | 20.00 | 26.67 | 42.33 | 64.33 |
| AM | 20.67 | 32.00 | 55.67 | 79.33 | 14.00 | 26.00 | 51.67 | 76.00 | 13.33 | 20.33 | 42.67 | 70.33 | 13.33 | 20.33 | 35.67 | 59.67 |
| CAM | 22.67 | **38.00** | **63.00** | **90.67** | 17.00 | 27.00 | **63.33** | **92.00** | 16.33 | 25.33 | 47.67 | **92.00** | 15.33 | 24.00 | **44.00** | **71.67** |

Table 11: **MRTA - TAPTC** Group 1: Average computation time (in milliseconds) to generate the entire solution for each scenario.

| | R25A2 | R25A3 | R25A5 | R25A7 | R50A2 | R50A3 | R50A5 | R50A7 | R75A2 | R75A3 | R75A5 | R75A7 | R100A2 | R100A3 | R100A5 | R100A7 |
|---|---|---|---|---|---|---|---|---|---|---|---|---|---|---|---|---|
| | | | | | | | Avg. Computation Time | | | | | | | | | |
| EILS | 19.00 | 102.67 | 1037.00 | 4221.33 | 181.33 | 517.00 | 1522.00 | 2914.67 | 90.00 | 248.67 | 972.33 | 5194.67 | 137.00 | 42.33 | 950.67 | 3329.00 |
| BiG-MRTA | 53.34 | 115.08 | 207.37 | 377.10 | 51.31 | 87.73 | 255.50 | 437.91 | 58.78 | 98.85 | 227.58 | 520.07 | 75.64 | 190.04 | 406.07 | 563.80 |
| AM | 103.81 | 96.44 | 96.75 | 95.22 | 95.65 | 94.90 | 95.05 | 105.27 | 94.81 | 111.42 | 133.90 | 107.05 | 99.81 | 101.72 | 129.88 | 97.53 |
| CAM(k=9) | 94.36 | 95.34 | 93.97 | 96.91 | 94.58 | 94.35 | 95.32 | 95.29 | 94.15 | 94.31 | 97.44 | 94.89 | 93.42 | 94.55 | 95.10 | 93.94 |

### F.6 IMPACT OF NEIGHBORHOOD SIZE FOR CAM ENCODER

Tables 12 and 13 compare the performance of CAM models with varying neighborhood size ($k$) in the encoder, based on the average completion rate (average of the 3 samples) for all the scenarios (all

values of $A$ and $R$ for the two groups). The impact of the neighborhood size $k$ (during encoding) is more evident on the performance of Group 1 test cases which has tasks with more number of tight deadlines. As shown in Table 12, the completion rate of CAM (with $k = \{6, 9, 12\}$) are almost comparable, while the performance of CAM with $k = 3$ is significantly lower than the other models. Lower the neighborhood size, smaller will be the local structural information learned, which could result in performance loss. However, increasing neighborhood size beyond a point may not necessarily improve performance. The average epoch time for training for the models with $k = \{3, 6, 9, 12\}$ are $11, 13, 14.3$, and $15$ minutes respectively. Figure 6 shows the learning curve for training all the CAM models and AM model.

Table 12: **MRTA - TAPTC** Group 1: Comparison of performance (average completion rate) of CAM models with varying neighborhood size ($k$). Higher the better.

| | Avg. Completion Rate | | | | | | | | | | | | | | | |
|---|---|---|---|---|---|---|---|---|---|---|---|---|---|---|---|---|
| | R25A2 | R25A3 | R25A5 | R25A7 | R50A2 | R50A3 | R50A5 | R50A7 | R75A2 | R75A3 | R75A5 | R75A7 | R100A2 | R100A3 | R100A5 | R100A7 |
| CAM_K_3 | 20.67 | 32.00 | 55.67 | 79.33 | 14.00 | 26.00 | 51.67 | 76.00 | 13.33 | 20.33 | 42.67 | 70.33 | 13.33 | 20.33 | 35.67 | 59.67 |
| CAM_K_6 | 23.00 | 37.33 | 61.67 | **91.00** | **17.33** | **27.67** | 61.00 | **92.00** | 15.67 | **25.33** | 45.33 | 91.00 | 14.67 | **24.33** | 41.67 | 71.33 |
| CAM_K_9 | 22.67 | **38.00** | **63.00** | 90.67 | 17.00 | 27.00 | 63.33 | **92.00** | 16.33 | **25.33** | 47.67 | 92.00 | **15.33** | 24.00 | **44.00** | **71.67** |
| CAM_K_12 | **24.00** | 37.67 | 62.33 | **91.00** | **17.33** | 27.00 | **63.67** | 91.33 | **16.67** | **25.33** | 45.67 | **92.33** | 15.00 | 23.67 | 43.00 | 69.67 |

Table 13: **MRTA - TAPTC** Group 2: Comparison of performance (average completion rate) of CAM models with varying neighborhood size ($k$). Higher the better.

| | Avg. Completion Rate | | | | | | | | | | | | | | | |
|---|---|---|---|---|---|---|---|---|---|---|---|---|---|---|---|---|
| | R25A2 | R25A3 | R25A5 | R25A7 | R50A2 | R50A3 | R50A5 | R50A7 | R75A2 | R75A3 | R75A5 | R75A7 | R100A2 | R100A3 | R100A5 | R100A7 |
| CAM_K_3 | 47.33 | 65.33 | 100.00 | 100.00 | 29.00 | 64.67 | 100.00 | 100.00 | 23.67 | 39.67 | 100.00 | 100.00 | 20.33 | 36.33 | 76.67 | 100.00 |
| CAM_K_6 | **54.33** | 76.67 | 100.00 | 100.00 | 40.33 | **76.67** | 100.00 | 100.00 | 30.00 | **58.33** | 100.00 | 100.00 | 25.67 | 42.67 | 100.00 | 100.00 |
| CAM_K_9 | 53.00 | **78.67** | 100.00 | 100.00 | **45.67** | 74.33 | 100.00 | 100.00 | 31.00 | 55.00 | 100.00 | 100.00 | **29.33** | **45.33** | 99.67 | 100.00 |
| CAM_K_12 | 54.00 | **78.67** | 100.00 | 100.00 | 40.33 | 75.67 | 100.00 | 100.00 | **32.00** | 54.67 | 100.00 | 100.00 | 27.33 | 44.33 | 96.00 | 100.00 |

# G   FURTHER DETAILS ON CAPACITATED VEHICLE ROUTING PROBLEM

## G.1   FORMULATION OF CAPACITATED VEHICLE ROUTING PROBLEM (CVRP)

The vehicle routing problem here we considered is a capacitated vehicle routing problem (CVRP), where a vehicle is required to deliver packages to a number of locations $N$. Each task is designated an index from 1 to $N$. We also consider a depot with id as 0. Each location has a demand $c_i$ on the number of packages where $i \in [1, N]$, and the vehicles has a constraint for the maximum number of packages $C$ it can carry, such that $c_i < D$. We assume that each package is of the same size. The vehicle is required to create multiple routes visiting different locations to deliver the packages. The vehicle starts from a depot, has a maximum capacity for the number of packages, and can have multiple routes to deliver all the packages satisfying the demands in every location. Here we assume that the vehicle can return to the depot for refilling to maximum capacity before starting a new route. In this experiment we do not consider split delivery where the demand of a location is fulfilled partially during a route, and then completed in another route. The ILP formulation for CVRP can be represented as:

$$\min \ f_{\text{cost}} = \Sigma \delta_j, j \in [1, R] \tag{20}$$

$$\text{subject to} \quad C_{t+1} = \begin{cases} \max(0, C_t - c_i) & i \notin V \\ D, & i = 0 \end{cases} \tag{21}$$

where $R$ is the number of routes, $\delta_j$ is total distance travelled in route $j$, $C_t$ is the available capacity at a time $t$, $V$ is the set of locations visited. The node encoding and the context encoding are modified for CVRP (as explained in appendix G.1.1 and G.1.2) for both CAM and AM. Both CAM and AM are trained for the scenarios with 100 locations and tested on the unseen scenarios with varying number of locations ranging from 50 to 1,000. The experimental details of this comparative study are given in Appendix G.2.

### G.1.1   ENCODING FOR CVRP

Except for the node properties, all the other steps for computing the node embeddings is same for CVRP as compared to MRTA for CAM and AM. The node properties associated with the CVRP

includes the $x$ coordinate, $y$ coordinate, and the demand $c_i$ for each location. Therefore each node can be represented as $d_i = [x_i, y_i, c_i]$, where $x_i$, $y_i$, and $c_i$ are the $x$ coordinate, $y$ coordinate, and capacity respectively for node $i$. For the encoding, $d_i$ for CVRP will be used in equation 22 for CAM.

$$F_i^d = W^d d_i^T + b_d \tag{22}$$

### G.1.2 Context encoding

The context serves the same purpose for which is represent the current state. For CVRP the current location and the remaining capacity as the context, for CAM and AM.

### G.2 Training details for CVRP

The training procedure for CVRP follows algorithm 1 in appendix D with the only change being in the calculation of the cost. Equation 20 is used in the algorithm 1 to compute the cost. Table 14 shows the different parametric setting for training CAM and AM for CVRP. CAM and AM models were trained for 100 tasks and tested on CVRP with 5 different task sizes (50, 100, 200, 500, 1000). Figure 7 shows the learning curve for training CAM and AM.

Table 14: **CVRP**: Training algorithm settings for CAM and AM for CVRP

| Details | AM | CAM |
|---|---|---|
| Algorithm | REINFORCE | REINFORCE |
| Baseline | Rollout | Rollout |
| Epochs | 100 | 100 |
| # of tasks | 100 | 100 |
| Training samples | 20000 | 20000 |
| Baseline samples | 1000 | 1000 |
| Optimizer | Adam | Adam |
| Learning step size | 0.0001 | 0.0001 |
| Training frequency | 100 SAMPLES | 100 SAMPLES |

### G.3 Baseline details for CVRP

**Lin-Kernighan heuristics (LKH3)**: We performed a single run with a maximum number of trails as 10000.

**AM**: Same as that of MRTA-Multi-UAV flood response and MRTA-TAPTC.

**Google OR tools:** The first solution strategy algorithm used was *PATH CHEAPEST ARC* and the local search algorithm was *Simulated Annealing*.

### G.4 More details on CVRP dataset

Table 15: The capacity of the vehicle for different test scenarios

| # of Tasks | Capacity (C) |
|---|---|
| 50 | 40 |
| 100 | 50 |
| 200 | 100 |
| 500 | 250 |
| 1000 | 500 |

The dataset used for training CVRP consists of scenarios with 100 locations and one depot. The $x$ and $y$ coordinates of the locations (including the depot) are randomly generated from a uniform distribution within the limits [0, 1]. The demand for all task locations will be a random integer from a uniform distribution between [1,9], with depot assigned a 0 demand. The vehicle capacity $(C)$

for a scenario with 100 locations, is considered as 50. The dataset used for testing (to analyze both generalizability and scalibility) has the same limits as explained above. The assumed capacity of the vehicle for test scenarios of different number of locations are shown in Table 15.

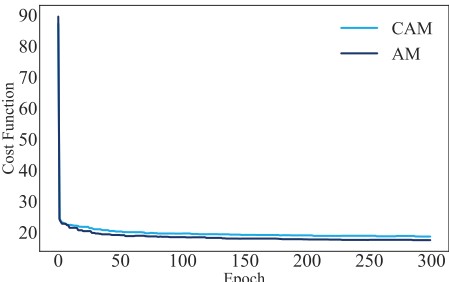

Figure 7: **CVRP:** Learning curve of CAM and AM (trained on problem size with 100 locations).

## H    LIMITATIONS OF CAM

This work is implemented for a fixed number of nodes but can be easily extended for cases where nodes are determined dynamically. The impact of the learning algorithm parameters such as the learning rate, training frequency or training batch size, etc. is not analyzed. Parametric analysis of the learning algorithm, as well as the implementation of more recent state-of-the-art RL algorithms (e.g., PPO), can be considered as other directions of future work with CAM. The current learning framework for CAM has been implemented with a greedy approach for decision-making. The performance can be improved by adopting an epsilon greedy approach. In real-world settings, it is possible that two robots want to make decisions at the same time, which might cause them to visit the same location. This is very rare and there are various mitigations to address this issue. For example, while robots are moving toward their selected task locations, they can check if their decision is conflicting with another robot based on recent information. If there is conflict, the robot with the worst time can cancel its current task and select a new task.

## I    ABBREVIATIONS:

Table 16: Abbreviations used in this paper

| | |
|---|---|
| CAM | Covariant-Attention based Model |
| CCN | Covariant Compositional Networks |
| CO | Combinatorial Optimization |
| CVRP | Capacitated Vehicle Routing Problem |
| EILS | Enhanced Iterated Local Search |
| GNN | Graph Neural Network |
| ID | In-schedule Dependencies |
| ILP | Integer Linear Programming |
| ILS | Iterated Local Search |
| INLP | Integer Non-Linear Programming |
| MDP | Markov Decision Process |
| MHA | Multi-Head Attention |
| MR | Multi-Robot |
| MRTA | Multi-Robot Task Allocation |
| mTSP | multi-Traveling Salesman Problem |
| RL | Reinforcement Learning |
| SR-ST | Single-Robot task Single Task robot |
| TAPTC | Task Allocation Problem with Time and Capacity |
| TSP | Traveling Salesman problem |
| UAV | Unmanned Aerial Vehicle |
| VRP | Vehicle Routing Problem |

