# OpenReview forum: "Learning to Solve Multi-Robot Task Allocation with a Covariant-Attention based Neural Architecture"
_ICLR.cc/2022/Conference — ICLR 2022 Submitted_

### Official Review · Reviewer_tBqb · 2021-11-02

**Correctness:** 3
**Technical Novelty And Significance:** 2
**Empirical Novelty And Significance:** 3
**Recommendation:** 5
**Confidence:** 2

**Main Review:**

This paper considers a very real-world problem, and there are many details in the context. The reviewer is not very familiar with this sub-area. As an application paper, it might be acceptable that the proposed method is a combination of some commonly used architectures or algorithms, such as multi-head attention, encoder-decoder and RL algorithms. In general, the technical novelty is not enough, although the empirical results are significant. Specifically, the reviewer is not sure what insights can this paper bring to AI community.

**Summary Of The Paper:**

This paper considers the multi-robot task allocation problems. To address the limitations of existing studies, such as real-world constraints, larger-sized problems and generalizations, this paper proposed a learning architecture, Covariant Attention-based Mechanism. They further conduct adequate evaluations and the results have shown great improvements over the state-of-the-art methods.



**Summary Of The Review:**

An application paper with adequate evaluations.

---

> ### Author Response · Authors · 2021-11-18
> **As an application paper, it might be acceptable that the proposed method is a combination of some commonly used architectures or algorithms, such as multi-head attention, encoder-decoder and RL algorithms. In general, the technical novelty is not enough, although the empirical results are significant. Specifically, the reviewer is not sure what insights can this paper bring to AI community.**
>
> We thank the reviewer for appreciating the real-world applicability of the method. We would like to add that the CAM architecture is applicable to different MRTA and VRP problems, which themselves are representative of a wide range of multi-agent system problems (possibly generalizing to an even wider range of CO problems). Thus the contributions and capabilities of CAM are not necessarily application-specific. For demonstration, we are evidently limited to using a few different case studies such as the MRTA-flood response, the benchmark MRTA-TAPTC, and the benchmark CVRP problem, and this scope of numerical experiments is at par with other well-known recent papers in the area.
> In addition, we also demonstrate how the MRTA problem can be formulated as an MDP over graphs, the design of the context portion could motivate the adoption of this formulation and associated GNN architecture in different MRTA and VRP type problems. In its current form, this approach is particularly suited for problems where time-critical feasible planning decisions are needed. Examples include disaster response and post-disruption reconfiguration of critical infrastructure networks (that can be modeled as multi-agent systems).
> In terms of novelty and insights for the AI community, I would like to point out the following key observations and new knowledge contributed thereof:
>
> 1) We are the first to explore that covariant compositional encoding of (task) graphs can provide better performance in learning to solve CO problems through RL, in comparison to the popular attention mechanism based encoding. The premise for this performance gain is the ability to better capture local structural information of the task graph; this characteristic is now discussed, with supporting illustration, in Appendix E2. While a rigorous theoretical analysis of when and how capturing local structural information (about node neighborhoods) helps policies perform better in providing CO solutions is not currently available, our empirical results provide significant evidence of its benefit over the attention mechanism across the MRTA and CVRP problems.
>
> 2) We show the feasibility of extending multi-head attention mechanisms (MH-A), serving as a decoder, from single agent to multi-agent problems, by incorporating a specially designed “context” portion inside the graph neural net. The decoder is fed information both from the encoder and this context portion. This context portion represents the robot’s self and peers state information as a learnable feature vector as explained in Section 3.2. Note that Kool et al.’s work only demonstrates MH-A capability in single-agent problems.
>
> 3) We show that this encoder-context-decoder architecture can produce policies that readily scale to problems that are larger in size than those used in training. This has tremendous implications in making such AI-based decision-making feasible on real-world problems. Many real-world problems will require such AI models to be trained over simulation or game environments (e..g., robotic simulators), where each episode has a significant run-time. Hence, training on larger problems might be computationally prohibitive even when significant offline computing resources are available, which is for example a well-known challenge in the multi-robot learning community. In such situations, the ability to train on smaller-sized problems, while still providing acceptable performance on larger problems could be critical to the adoption of AI-based decision-making.
>
> [1] Kool, W., Van Hoof, H., & Welling, M. (2019). Attention, learn to solve routing problems! 7th International Conference on Learning Representations, ICLR 2019.

---

### Official Review · Reviewer_Yr3w · 2021-11-02

**Correctness:** 2
**Technical Novelty And Significance:** 2
**Empirical Novelty And Significance:** 3
**Recommendation:** 3
**Confidence:** 4

**Main Review:**

Strengths:

+ The idea of using learning methods to solve a SR-ST MRTA problem is interesting.

+ Discussion of the technical approach is clear.

Weaknesses:

- The theoretical novelty is not high. The decoder and the training algorithm REINFORCE are similar or identical to the method by Kool et al (2019). The main novelty is to add constraints to the optimization objective of MRTA. However, it is not clear what new challenges are caused by adding the constraints? Theoretically, why does these constrains make the optimization problem more challenging to solve?

- The paper states that "this MRTA problem is adopted from Ghassemi et al (2019) with the modification of removing payload". However, the optimization problem introduced in Ghassemi et al (2019) includes other constraints that are ignored by the paper. Other than payload, why other constraints are removed?

- What is the meaning of t? The paper explains that "the transition is an event-based trigger", so it seems that t means the index of events (e.g., in Figure 1). In this case, when an event happens and all robots choose their tasks, how to guarantee that no multi-robots select the same task?

- The terms generalizability and scalability are used throughout the paper. But they are not clearly defined, and in some places, they are used interchangeably, e.g., "generalizing to problem scenarios that are larger in size".

- The experimental results are not clearly presented, especially the evaluation metrics. Besides the issue of clearly defining generalizability and scalability, can the authors provide the single value metric to quantify generalizability and scalability?

- What are the task completion time of each approach for the testing scenarios in Table 1 and Table 2?

- Without a centralized system, assuming each robot knows the information from all other robots in the swarm is not practical for real-world swarm applications.

- Minor: The variable t_i is used for both deadline and event/timestep index.


**Summary Of The Paper:**

This paper introduces a deep graph policy learning method that chooses tasks for each individual robot in a swarm thus addressing the problem of single-task robot and single-robot task (SR-ST) multi-robot task allocation (MRTA). The proposed method is based on the work by Kool et al (2019) with the main theoretical extensions on (1) adding constrains to the optimization objective of MRTA and (2) improving the encoder.

**Summary Of The Review:**

Although the idea of using learning to address MRTA problems is interesting, the paper is preliminary and will need significant improvements on novelty justification, terminology definition, and experimental result presentation.

---

> ### Author Response · Authors · 2021-11-18
> **Without a centralized system, assuming each robot knows the information from all other robots in the swarm is not practical for real-world swarm applications.**
>
> In many real-world applications, such as last-mile delivery, the full observability of tasks and robots in the time-scale of seconds is feasible [1]. The communication bottleneck is a key issue in swarm control algorithms when a high frequency of decision-making and data sharing is required. However, in our MRTA application, each robot is making a decision every few seconds or minutes. This means that one can use multi-hop communication or other long-range communication technology to aggregate information about tasks and other robots in a timely fashion [1, 2]. However, we do agree that the wider range of problems that can be solved by CAM does include some application instances (as mentioned above) where partial observability effects are significant. In such cases, the proposed CAM architecture can also be implemented to consider partial observability, where instead of operating over an MDP, the policy model will operate over a POMDP. For this purpose, the context portion of CAM needs to be updated by including the information of the neighbors (robots in the locality of the decision-making robot), and/or probabilistic predictions over the information of other peer robots that are out of range. However, including partial observability is out of scope for this paper.
>
> [1] Kagawa, T., Ono, F., Shan, L., Miura, R., Nakadai, K., Hoshiba, K., Kumon, M., Okuno, H. G., Kato, S., & Kojima, F. (2020). Multi-hop wireless command and telemetry communication system for remote operation of robots with extending operation area beyond line-of-sight using 920 MHz/169 MHz. Advanced Robotics. https://doi.org/10.1080/01691864.2020.1760934
>
> [2] Mehta, Vaibhav Kumar, and Filippo Arrichiello. "Connectivity maintenance by robotic Mobile Ad-hoc NETwork." arXiv preprint arXiv:1312.2526 (2013).

---

> ### Author Response · Authors · 2021-11-18
> **What are the task completion time of each approach for the testing scenarios in Table 1 and Table 2?**
>
> Mission completion time can be defined as the duration that all the available tasks have been visited by robots; in the presence of task deadlines (as is the case here), some tasks might expire as well over time (without ever being allocated to any robot), which is why “task completion rate” is considered to be a more apt representation of a performance in real-world applications such as disaster response. More importantly, note that for the MRTA problem being solved here, the composite cost or reward function includes both task completion rate (i.e., the fraction of tasks completed) to be maximized, and total distance traveled by robots, to be minimized. As a result, any learning or non-learning method acting on this cost function is not necessarily seeking to minimize mission completion time. Therefore, it is not meaningful or fair to report the mission completion time of the different methods for this study.

---

> ### Author Response · Authors · 2021-11-18
> **The experimental results are not clearly presented, especially the evaluation metrics. Besides the issue of clearly defining generalizability and scalability, can the authors provide the single value metric to quantify generalizability and scalability?**
>
> It is not clear to us what clarity is missing from the presentation of the results. Results are reported in terms of “cost function” associated with the respective problem (e.g., MRTA and CVRP) and the computing cost of evaluating decisions (the entire sequence of tasks). These metrics are quite standard in the MRTA, CVRP, and related multi-agent system literature, and is widely used across learning and non-learning paradigms.
> One very common way of looking at generalizability is looking at the test performance of a policy over unseen scenarios, which is what we use here. Comparing training and test performance could also shed light on the architecture’s ability to generalize. Similarly, one way of looking at scalability is how the performance and computing cost changes as the size of the problem increase. Interestingly, here we study scalability on problem scenarios that are even larger than those scenarios used for training (in terms of # task and robots). While we agree that a single scalar measure each of generalizability and scalability (that does not depend on cross-method comparisons) will be helpful for the analyses of solution approaches in the multi-agent systems and CO community, development of such metrics is not within the intended aims of this paper.

---

> ### Author Response · Authors · 2021-11-18
> **The terms generalizability and scalability are used throughout the paper. But they are not clearly defined, and in some places, they are used interchangeably, e.g., "generalizing to problem scenarios that are larger in size".**
>
> We have updated section 5.1 to clearly define generalizability and scalability as being analyzed in this paper. This changed portion is marked in blue font in the revised manuscript.

---

> ### Author Response · Authors · 2021-11-18
> **What is the meaning of t? The paper explains that "the transition is an event-based trigger", so it seems that t means the index of events (e.g., in Figure 1). In this case, when an event happens and all robots choose their tasks, how to guarantee that no multi-robots select the same task?**
>
> The term t is the time when an event occurs, i.e., when a robot reaches a location or completes its task. The index of the events is given by the subscript of t. For example, the time of the 2nd event is t_2. In the simulation, it is not possible to have conflicting decisions, as the sharing of information is almost instantaneous. However, in real-world settings, it is possible that two robots want to make decisions at the same time, which might cause them to visit the same location. This is very rare and there are various mitigations to address this issue. For example, while robots are moving toward their selected task locations, they can check if their decision is conflicting with another robot based on recent information. If there is conflict, the robot with the worst time can cancel its current task and select a new task. This has been clarified in the revised manuscript in Section 3.

---

> ### Author Response · Authors · 2021-11-18
> **The paper states that "this MRTA problem is adopted from Ghassemi et al (2019) with the modification of removing payload". However, the optimization problem introduced in Ghassemi et al (2019) includes other constraints that are ignored by the paper. Other than payload, why other constraints are removed?**
>
> This seems to be a misunderstanding possibly due to how the case study was worded in the original manuscript. This has now been fixed. Section 2.2 of the revised manuscript now clearly states that CAM and AM are being used here to solve the multi-UAV flood response problem including all of the constraints as described in Ghassemi and Chowdhury 2019, 2021, with the exception of only the UAV payload capacity constraints. To further clarify, constraints associated with inter-robot conflicts, robot ferry range, and task deadlines are all being accounted for here.

---

> ### Author Response · Authors · 2021-11-18
> **The theoretical novelty is not high. The decoder and the training algorithm REINFORCE are similar or identical to the method by Kool et al (2019). The main novelty is to add constraints to the optimization objective of MRTA. However, it is not clear what new challenges are caused by adding the constraints? Theoretically, why does these constrains make the optimization problem more challenging to solve?**
>
> Kool et al. [1] adopt the transformer architecture, which is an attention-based encoder-decoder architecture and implemented in a variety of simple single-agent problems. This method performs poorly for more complex multi-agent problems involving additional constraints, as can be seen in our results from the MRTA problems (Tables 1-3). It’s also important to note that choice of REINFORCE as the RL training solver is quite common in the context of training GNNs with RL, and is in no way a contribution of Kool et al’s work. Given its simplicity, it’s often the first goto solver.
>
> Regarding the additional complexity or constraints in the MRTA problem: As stated in Section 1.1 in the original manuscript, for the studied SR-ST problem with “n” locations, ‘m’ robots, and assuming ‘h’ number of routes, the complexity for the exact integer (non)linear programming (ILP) is O(n^3m^2h^2 ). The problem can be linear or non-linear depending on the nature of the application-driven cost function. This computational complexity is more significant than that of classical VRP or TSP problems that have been studied by Kool et al. [1].
> As a result, the cost of solving the problem online increases significantly with increasing number of tasks and robots, when using state-of-the-art non-learning based methods; to a point where such non-learning solutions (even with online MRTA methods) become prohibitive to be used (e.g., problems with 1000’s of tasks) when real-time decisions are desired. These characteristics are evident from our scalability analysis. This is also where the proposed learning based methods, CAM and AM shine (i.e., in terms of their online computing performance), and on top of that, CAM provides significantly better cost function values compared to AM in almost all scenarios that we studied across different numerical experiments.
> Coming back to the question of novelty, there are three key contributions in this paper 1) Incorporating a new encoder, inspired by the Covariant Compositional Network (CCN), to increase the ability of the architecture to capture better local structural information of the nodes/tasks than an attention-based encoder (Kool et al. [1]); we are not aware of any other work besides our which has explored the capacity of CCN is helping solve combinatorial optimization (CO) problems of the type of TSP, VRP and MRTA, and hence the resulting findings clearly adds new knowledge to the emerging domain of “learning to solve CO problems”; 2) Introducing the context portion to capture the robots’ states in manner such that it remains agnostic to the number of robots, thereby allowing scaling to larger problems without network structural issues; and 3) Providing formulation of the MRTA problem as an MDP that can be solved in a decentralized and asynchronous manner.
>
>
> [1] Kool, W., Van Hoof, H., & Welling, M. (2019). Attention, learn to solve routing problems! 7th International Conference on Learning Representations, ICLR 2019.

---

### Official Review · Reviewer_uLHX · 2021-11-03

**Correctness:** 3
**Technical Novelty And Significance:** 3
**Empirical Novelty And Significance:** 3
**Recommendation:** 6
**Confidence:** 3

**Main Review:**

The strength of the paper is that it proposed a new and effective method for MRTA problems. To interpret the task as an MDP is not new. However, the architecture based on the multi-head attention-based encoder-decoder network is interesting.
The weakness of the paper is about the generality of the theory and the evaluation. In contrast with the general discussion and argumentation in the Introduction, the proposed method contains many specific formulation and network architectures (e.g., equations (1) and (2), and network architecture shown in Figures). How much generality does the proposed framework have? Which part actually did contribute to the performance? Ablation studies are expected to clarify these points. The evaluation was conducted in terms of generalizability, scalability, completion rate, and computation time. However, apparently, computation time and quality of solution have a trade-off relationship. The discussion or fair evaluation about the trade-off should be mentioned. For example, in a standard optimization problem, we can get a little worse solution by limiting the computational time (i.e., terminating the iterative algorithm). It is clearly shown that the proposed method is better in computation time. However, the performance considering the trade-off should be fairly discussed.


<Minor comments>
1. Deterministic assumption:
They assume the dynamics are deterministic. However, their theory is based on MDP. Therefore, the approach seems to be naturally applied to a probabilistic environment as well. I suggest the authors mention this point.

2. The first sentence of 2.1
The sentence starting with "The MRTA problem can…" may be inaccurate. (A graph itself is not a "problem.")

3. Too small figures and tables.
Many figures (e.g., 1) and tables (e.g., 3 and 4) are too small to read. Please make them larger and improve the readability.

4. Highlight color
Table 3 and some tables in Appendix have a green background as a highlight. If you use the dark colors (e.g., green, blue and purple) as "highlight," the readability becomes significantly worse.
Please use other styles (e.g., underline or bold style).

5. Topline
In Table 1, BiG-MRTA is shown as a topline. It's better to use a different style that allows readers to easily distinguish the topline from baselines. For example, you can use a double line between BiG-MRTA and AM.
Also, it helps readers grasp the information to give underline or bold to the best value.

6. Qualitative evaluation
Only quantitative results are shown. Providing qualitative evaluations may help readers understand the characteristic of the proposed methods.




**Summary Of The Paper:**

This paper proposes a new method, including neural network architecture, for solving time-constrained multi-robot task allocation (MRTA) problems. The proposed approach models the target problem as a Markov Decision Process (MDP) over graphs and use Reinforcement Learning (RL) methods to solve the problem.
The proposed learning architecture is called Covariant Attention-based Mechanism (CAM).
The architecture is shown to have better performance than an existing state-of-the-art encoder-decoder method regarding task completion, cost function, and scalability. Though the performance is still lower than non-learning-based baseline methods, i.e., BiG-MRTA, the computational cost is significantly smaller than the baselines.


**Summary Of The Review:**

This paper proposes a new learning architecture which is called Covariant Attention-based Mechanism (CAM), to solve time-constrained multi-robot task allocation (MRTA) problems using a learning-based approach. The method works in an appropriate manner, and empirical results show the advantage of the method. Also, the approach itself is based on a modern approach and is worth studying.
As a whole, the paper is a good paper involving meaningful new information.

---

> ### Author Response · Authors · 2021-11-18
> **Qualitative evaluation Only quantitative results are shown. Providing qualitative evaluations may help readers understand the characteristics of the proposed methods.**
>
> For qualitative evaluation, we have performed an ablation study (in Appendix E2), to understand the importance of the CCN based encoding and the MHA-based decoding. For the encoder, the CCN based encoding was disabled in the encoder and was replaced with a simple feedforward network encoding, with the decoder being the same. For the decoder, the MHA-based decoding was disabled for the decoder and was replaced with a simple feedforward network and a softmax layer, with the encoder not changed.
> In both cases, there was a significant decrease in the performance (table 8) for all the scenarios, with the maximum dip in the completion rate being 19.8% and 13.4% for the first and the second case, respectively.
>
> Compiling the results from the first case of the ablation study for the encoder and with the comparison with AM, we posit that the CCN based encoding, which is able to aggregate local node neighborhoods while remaining agnostic to node ordering, aided in better policies.

---

> ### Author Response · Authors · 2021-11-18
> **Topline In Table 1, BiG-MRTA is shown as a topline. It's better to use a different style that allows readers to easily distinguish the topline from baselines. For example, you can use a double line between BiG-MRTA and AM. Also, it helps readers grasp the information to give underline or bold to the best value.**
>
> As per the suggestion, we have made the topline for tables 1,2, and 5 as bold for better readability.

---

> ### Author Response · Authors · 2021-11-18
> **Highlight color Table 3 and some tables in Appendix have a green background as a highlight. If you use the dark colors (e.g., green, blue and purple) as "highlight," the readability becomes significantly worse. Please use other styles (e.g., underline or bold style).**
>
> We have removed the highlight color from tables 3, 9, 11, and 12,  and have highlighted the best result in each case in bold font.

---

> ### Author Response · Authors · 2021-11-18
> **Too small figures and tables. Many figures (e.g., 1) and tables (e.g., 3 and 4) are too small to read. Please make them larger and improve the readability.**
>
> The size of the figures has been increased for better readability. The font size of the tables has also been increased.

---

> ### Author Response · Authors · 2021-11-18
> **The first sentence of 2.1 The sentence starting with "The MRTA problem can…" may be inaccurate. (A graph itself is not a "problem.")**
>
> This sentence has been restated as below and highlighted in blue color.
>
> 	“The MRTA problem has a set of nodes/vertices V and a set of edges E that connect the vertices to each other, which can be represented as a complete graph G = (V, E).”

---

> ### Author Response · Authors · 2021-11-18
> **Deterministic assumption: They assume the dynamics are deterministic. However, their theory is based on MDP. Therefore, the approach seems to be naturally applied to a probabilistic environment as well. I suggest the authors mention this point.**
>
> We have added the following text to Section 3.1 and highlighted in blue.
>
> 		“It should be noted that the encoding mechanism here, can also be extended for a probabilistic scenario, for example, when the deadline  follows a probabilistic distribution.”

---

> ### Author Response · Authors · 2021-11-18
> **Regarding ablation studies, and trade-off between performance and computation time**
>
> n the revised manuscript, in Appendix E2,  we discussed the effect and importance of each component of the proposed architecture using an ablation study. The paper has been revised to clearly show how each component contributes to the final performance.
> For standard optimization methods, besides expensive computing time, there is a memory limitation. When the size of the MRTA problem is large, the amount of the required memory increases significantly. The learned model does not suffer from this limitation. In order to address this issue in the non-learning-based methods, there exist various works that propose heuristic-based methods to find approximated solutions in a timely manner [1]. One of these methods is EILS [2]. In this paper, as discussed in Section 5.2, EILS has been compared with CAM and AM. As shown in Tables 3 and 4, CAM and AM generate better or comparable solutions w.r.t. EILS for large problem sizes, while they are at least 10 times faster than EILS.
>
>
> [1] Mosteo, Alejandro R. and Luis Montano. “A survey of multi-robot task allocation.” (2010).
>
> [2] Mitiche, H., Boughaci, D., & Gini, M. (2019). Iterated local search for time-extended multi-robot task allocation with spatio-temporal and capacity constraints. Journal of Intelligent Systems. https://doi.org/10.1515/jisys-2018-0267

---

### Official Review · Reviewer_AMwa · 2021-11-03

**Correctness:** 3
**Technical Novelty And Significance:** 3
**Empirical Novelty And Significance:** Not applicable
**Recommendation:** 8
**Confidence:** 3

**Main Review:**

1. The paper is well written and well organized. The references are appropriate and the problem being addressed is very well explained.

2. One of the paper contributions is the modeling of the MRTA problem as a MDP over graphs. This formulation is important because it allows to propose the covariant attention-based neural architecture (CAM) to learn MRTA problems.

3. The CAM architecture is the main paper contribution. The CAM is basically an encoder-decoder architecture, it uses attention mechanisms in the encoder and in the decoder, it uses information of other agents (context), and it codifies spatial information using a Graph Neural Network. The author put nicely together all these different elements in the CAM, which is able learn appropriately MRTA problems.

4. The reported results are convincing. Case studies are presented in which the proposed CAM is compared with other learning based and non-learning based methods. In the reported experiments CAM obtained smaller errors (average cost ) and smaller processing time. CAM scales well with the number of tasks and the number of robots.

5. It could be good that authors could explain about the limitations of CAM. It can really be used in any CAM problem? Is it  easy to use? Is tit easy to model any CAM problem and to put as required by the CAM?


**Summary Of The Paper:**

This paper proposed a neural architecture for learning to solve multi-robot task allocation (MRTA) problems. The MRTA problem is modeled as a MDP, and then the so-called covariant attention-based neural architecture (CAM) is proposed. The main paper contribution is the CAM architecture.

Case studies are presented in which the proposed CAM is compared with other learning based and non-learning based methods. In the reported experiments CAM obtained smaller errors (average cost) and smaller processing time.


**Summary Of The Review:**

This is a novel paper. The proposed covariant attention-based neural architecture (CAM) is a novel architecture as well as the modeling of MRTA problems as MPDs. Moreover, the use of the CAM architecture allows solving MRTA problems in a better way than using tradition methods. MRTA scales well with the number of robots/agents and with the number of tasks.

---

> ### Author Response · Authors · 2021-11-18
> **Regarding limitations and ease of use**
>
> Limitation: This work is implemented for a fixed number of nodes but can be easily extended for cases where nodes are determined dynamically. The impact of the learning algorithm parameters such as the learning rate, training frequency or training batch size, etc. is not analyzed. Parametric analysis of the learning algorithm, as well as the implementation of more recent state-of-the-art RL solver(e.g., PPO), can be considered as other directions of future work with CAM. The current learning framework for CAM has been implemented with a greedy approach for decision-making. The performance can be improved by adopting an epsilon greedy approach. We have included the limitations of our work in Appendix H of the revised manuscript.
>
> Ease of use: CAM model can be implemented for a wide variety of multi-agent task allocation problems, by making the necessary changes in the encoder and the context portion. CAM can also be implemented on single-agent problems similar to [1].
>
> [1] Kool, W., Van Hoof, H., & Welling, M. (2019). Attention, learn to solve routing problems! 7th International Conference on Learning Representations, ICLR 2019.

---

### Official Review · Reviewer_8Xfc · 2021-11-03

**Correctness:** 3
**Technical Novelty And Significance:** 2
**Empirical Novelty And Significance:** 3
**Recommendation:** 6
**Confidence:** 3

**Main Review:**

**Strengths**

The paper tackles the important problem of MRTA that is typically a computationally expensive optimization problem. The proposed approach of learning heuristics to solve it quickly seems appealing. Since the optimization is over a graph, choosing a graph neural network architecture seems like a natural fit that scales for varying numbers of tasks and robots without retraining the model. The paper also presents a substantial evaluation comparing it to a SOTA non-learning baseline.

**Weaknesses**

The main weakness of the paper is that it is unclear from the results if the CAM policy is learning something meaningful. Without clearly showing this, it is difficult to accept the paper at this time.

Looking at Tables 1 and 2, it appears that even though CAM is faster than BiG-MRTA, the performance of the policy is substantially worse. Interestingly, this is especially pronounced when the robot to task ratios are low (i.e. 1:10). This suggests that the policy being learned is possibly myopic, i.e., robots greedily pick tasks that suffices when ratios are high (1:5 or higher).

I would like to see a qualitative result showing the actions chosen by CAM vs BiG-MRTA, especially in the low robot to task ratio regimes. This would help convince the reader that the policy being learned is indeed meaningful.

**Suggestions for improvement**

*Overall story*

The paper prescribes a very specific architecture for learning policies that solve MRTA problems. However, it does not offer much insight into why this approach is preferred over alternatives. For instance, an ablation study can be provided to show the relative importance of different components -- why do we need the attention layer as opposed to a simpler aggregation step in GNN, rounds of message passing in the graph, etc. Additionally, for readers unfamiliar with MRTA problems, the paper appears difficult to follow. The paper would benefit from providing more insight into why one needs to apply learning for this problem, why GNN architecture is natural, why RL over other paradigms such as imitation learning of offline solvers.

*Technical Points*

The paper assumes full observability, i.e., each robot has the full state of other robots. Practically, for real-world problems with communication bottlenecks, this seems unrealistic. It would be good to explain how this assumption can be relaxed for applications in practical settings. It was also unclear how the architecture scales to dynamically changing robots since the size of the context would vary, requiring retraining. Results in Section 5.2 claim that the same model scaled to a varying number of robots. It would be good to have a subsection in section 3 that explains how the varying context is handled.

The reward function defined seems very sparse, i.e., 0 for all states until all tasks are inactive. Historically, REINFORCE performs poorly in such settings, mostly doing random explorations till it sees high reward states enough times. It could be interesting to compare this against an approach that uses imitation learning to bootstrap RL. For instance, the results show that the non-learning baseline BiG-MRTA does quite well, outperforming the proposed approach on many trials. How well does this approach fair against a baseline that imitates the BiG-MRTA policy? How much does REINFORCE improve upon this policy by using this as a warm start?

Finally, the visited tasks are masked after prediction. By doing this, the model has no idea that it should not put all probability mass on visited tasks. This may result in random tie breaks. How would the approach compare to a method that takes the visited tasks as input to the CAM model?

*Formatting and style*

As presented, the paper appears dense and at times difficult to read. This is likely due to a lot of details pertaining to MRTA, architectural details, and result descriptions. These details can easily be moved to the appendix and the main body of the paper can focus on the higher-level interpretation of results (e.g. the proposed approach, though faster than big BiG-MRTA, actually performs poorer on rewards). I would also recommend minimizing the use of abbreviations or having a table of abbreviations for reference and using new line enumerations to further improve the readability of the paper.

**Summary Of The Paper:**

The paper proposes a graph learning approach for solving the multi-robot task allocation (MRTA) problem. It frames the problem as a Markov Decision Process (MDP) and trains a policy with a graph neural network architecture using REINFORCE. Results show that the proposed approach scales better compared to a non-learning baseline and is more accurate than a multi-headed attention (MHA) approach.

**Summary Of The Review:**

My overall recommendation is weak reject. While the paper tackles learning for a difficult optimization problem, it is not clear from the results whether it succeeded in learning something meaningful.

---

> ### Author Response · Authors · 2021-11-18
> **Formatting and style**
>
> We have moved the section “Computing time (Training and Execution)” (previously Section 5.1)  to Appendix D5.1, as well as the section “Design of Experiments & Learning Procedures” (previously Section 4.1) to Appendix D3. We have also added a table for abbreviations in Appendix I (Table 16)

---

> ### Author Response · Authors · 2021-11-18
> **Regarding masking**
>
> The approach where masking is disabled is worth exploring. Removing masks increases the training time. During actual implementation masking of visited and infeasible tasks was observed to usually result in better decision-making than when masking is disabled.

---

> ### Author Response · Authors · 2021-11-18
> **How well does this approach fair against a baseline that imitates the BiG-MRTA policy? How much does REINFORCE improve upon this policy by using this as a warm start?**
>
>  We thank the reviewer for the suggestion. In fact, this is also one of the future directions of our research. Unfortunately, we will not be able to include the imitation learning part, since it is beyond the scope of this paper, and would be challenging to include given the space limitations. Please refer to our earlier response w.r.t. How a hybrid imitation learning approach could further improve the performance of graph learning in solving such problems.

---

> ### Author Response · Authors · 2021-11-18
> **Regarding full observability, connectivity, and handling varying number of robots**
>
> Connectivity: In many real-world applications, such as last-mile delivery, the full observability of tasks and robots in the time-scale of seconds is feasible, as noted in various articles [1]. The communication bottleneck is a key issue in swarm control algorithms when a high frequency of decision-making and data sharing is required. However, in our MRTA application, each robot is making a decision every few seconds or minutes. This means that you can use multi-hop communication or other long-range communication technology to aggregate information about tasks and other robots [1, 2].
>
> Partial observability: The CAM architecture can also be implemented for partial observability. This can be achieved by modifying our context portion of CAM, by including just the information of the peer robots with which the robot taking the action can communicate at that time step. However, including partial observability is out of scope for this paper.
>
> Varying number of robots: The context portion of CAM has been designed such that it is agnostic to the number of robots. For example, the context portion for the MRTA problem contains the five following components 1) Current time,2) Current location of the robot taking, 3) Available range for the robot making a decision, 4) Current destination of the peer robots,  5) Available range for the peer robots. The current state of the robot taking decision (which 2 and 3) undergoes a linear transformation to produce a single feature vector. Similarly, the states of the robots also undergo a linear transformation, and the resultant feature vectors are aggregated to be represented as a single feature vector (as shown in figure 2 of the manuscript). This aggregation operation makes the CAM model agnostic to the number of robots. Due to space constraints in the original manuscript, this discussion has been added in Appendix D1, in the revised manuscript (highlighted in blue font).
>
>
> [1] Kagawa, T., Ono, F., Shan, L., Miura, R., Nakadai, K., Hoshiba, K., Kumon, M., Okuno, H. G., Kato, S., & Kojima, F. (2020). Multi-hop wireless command and telemetry communication system for remote operation of robots with extending operation area beyond line-of-sight using 920 MHz/169 MHz. Advanced Robotics. https://doi.org/10.1080/01691864.2020.1760934
>
>
> [2] Mehta, Vaibhav Kumar, and Filippo Arrichiello. "Connectivity maintenance by robotic Mobile Ad-hoc NETwork." arXiv preprint arXiv:1312.2526 (2013).

---

> ### Author Response · Authors · 2021-11-18
> **Regarding comments under 'Overall story'**
>
> It seems that our descriptions in the original paper had left opportunities for further clarity. We revised the paper to address these gaps. Specifically, in Section 1.1, we discuss the main motivation to use a learning-based method for solving MRTA problems. One of the key advantages of learning-based methods over non-learning-based methods is their low computational time for generating solutions for large problems during operation. Our results in Tables 3-5 support this claim. It is possible that the learning-based method might even provide better cost function (e.g., task completion rate) performance when problems present additional complexities such as non-linearities and mixed-integer decision spaces, which derail state-of-the-art online MRTA methods. However, this is not explicitly studied here, as the flood response, benchmark MRTA and CVRP problems we use mostly boil down to large integer linear (or mildly non-linear) programming problems
>
> The main design choices of our architecture are for the encoder and the decoder. The importance of a CCN based encoder is evident by comparing the performance with the AM method (that is partly derived from Kool et al. ) , which has an attention-based encoder. As per the suggestion of the reviewer, we have included the results of our ablation study in appendix E2, showing the relative importance of the CCN-based encoder and the MHA-based decoder.
>
> In Section 1.1, we now discuss why graph neural networks (GNN) work well as policy models for combinatorial optimization problems (including MRTA).
>
> Imitation learning provides a great opportunity to speed up learning by utilizing expert demonstration (e.g., from a sparse set of optimal solutions given by offline solvers). As a matter of fact that’s an ongoing portion of the current sponsored project under which this research has been performed. There is indeed an opportunity to bootstrap labels from say BiG-MRTA or even optimal ILP solutions (at least for small problems), and construct a hybrid imitation learning process that minimizes deviations from expert policies (over sparse set of scenarios) while further generalizing across a wider range of scenarios in training. Frameworks such as GAIL [1] can be extended to work with graph neural nets to implement this process. However, exploring such a hybrid imitation learning is not within the scope of this paper, but is planned as future work under our ongoing research project. More importantly, the current RL method can be readily extended to such a hybrid imitation learning setting while continuing to use the same encoder-decoder architecture (as in CAM) which is the main contribution of this paper.
>
> [1] Ho, J., & Ermon, S. (2016). Generative adversarial imitation learning. Advances in Neural Information Processing Systems.

---

> ### Author Response · Authors · 2021-11-18
> **Regarding weakness**
>
> While the policies learnt by CAM could be myopic to some extent, they donot necessarily simply provision greedy feasible decisions To clearly demonstrate that CAM and AM are learning meaningful policies, we added a randomized myopic baseline, called Feasibility-preserving Random-Walk (Feas-RND) [1], where each robot randomly chooses tasks from among feasible tasks. The results for Feas-RND have been added to tables 1,2 and 7, and with further discussion in Section 5.1. CAM performs clearly better than Feas-RND in terms of the cost function across all cases studied in our generalizability and scalability analysis. CAM’s superiority to a greedy randomized highly-myopic task selection approach (like Feas-RND)  is further evident from its significantly superior task completion rate across these cases, which is now summarized in Table 7 in Appendix E.1
>
> [1] Ghassemi, P., & Chowdhury, S. (2022). Multi-robot task allocation in disaster response: Addressing dynamic tasks with deadlines and robots with range and payload constraints. Robotics and Autonomous Systems. https://doi.org/10.1016/j.robot.2021.103905

---

> ### Comment · Reviewer_8Xfc · 2021-12-01
> **Response to Authors**
>
> Thank you for the detailed response. I appreciate the additional evaluation and incorporation of suggestions. While it would still be insightful to see a qualitative result showing the actions chosen by CAM vs BiG-MRTA, the addition of the random baseline seem to indicate that the policy is learning something meaningful. As a result, I have updated my rating to weak accept.

---

### Author Response · Authors · 2021-11-18
**General comment from the authors**

We appreciate the thoroughness of the review process and thank the reviewers for their time and guidance. As per the recommendation of the reviewers, we have added more results, on the importance of different components of CAM using an ablation study, as well as for qualitative analysis, by comparison with a randomized myopic decision-making strategy called Feas-RND. The updates are highlighted in blue-colored font in the revised manuscript. We are posting a version of the revised manuscript now (November 18), that includes all revisions that are currently marked in our response below. This revised manuscript will get replaced with further updated versions as we continue to address further comments from the reviewers  (if any)  over the next few days.

---

### Decision · Program_Chairs · 2022-01-20

**Decision:**

Reject

**Comment:**

The paper considers the problem of solving time-constrained multi-robot task allocation (MRTA) problems. Formulating the problem as a Markov decision process (MDP), the paper proposes Covariant Attention-based Mechanism (CAM), a graph neural network-based policy that can be trained to solve MRTA problems via standard RL methods. The encoder adapts the covariant compositional network to improve generalizability, while the decoder extends a recent combinatorial optimization architecture to the multi-agent optimization domain. Experimental results demonstrate that CAM outperforms an encoder-decoder baseline in terms of task completion, generalizability, and scalability, while also providing greater computational efficiency than non-learning baselines.

The paper considers an important topic---multi-agent task allocation is an interesting and challenging combinatorial optimization problem. The proposed CAM architecture adapts existing components in an interesting way and seems sensible for the MRTA domain. The reviewers initially raised concerns regarding the conclusions that can be drawn from the experimental evaluation, the significance of the algorithmic contributions, as well as the motivation for the proposed approach. The authors made a concerted effort to address these concerns through the addition of new experimental evaluations (e.g., comparisons to a myopic baseline and ablation studies), updates to the text, and detailed responses to each reviewer. Unfortunately, only one reviewer responded and updated their review (increasing their score). In light of this, the AC also reviewed the paper. The AC agrees with the strengths identified by the reviewers (including those noted above) and with the contributions provided by the additional evaluations. However, the paper remains unnecessarily dense, while at the same time not being self-contained (e.g., the new experimental results are relegated to the appendix rather than appearing in the main text). The paper would also benefit from a more concise motivation for learning-based solutions to MRTA and a clearer discussion of the paper's contributions.